# Talin and vinculin combine their activities to trigger actin assembly

Hong Wang [1,2,3], Rayan Said[1,4], Clémence Nguyen-Vigouroux[1,4], Véronique Henriot[1], Peter Gebhardt[2], Julien Pernier[1], Robert Grosse [2,3] & Christophe Le Clainche [1]✉

Focal adhesions (FAs) strengthen their link with the actin cytoskeleton to resist force. Talin-vinculin association could reinforce actin anchoring to FAs by controlling actin polymerization. However, the actin polymerization activity of the talin-vinculin complex is not known because it requires the reconstitution of the mechanical and biochemical activation steps that control the association of talin and vinculin. By combining kinetic and binding assays with single actin filament observations in TIRF microscopy, we show that the association of talin and vinculin mutants, mimicking mechanically stretched talin and activated vinculin, triggers a sequential mechanism in which filaments are nucleated, capped and released to elongate. In agreement with these observations, FRAP experiments in cells co-expressing the same constitutive mutants of talin and vinculin revealed accelerated growth of stress fibers. Our findings suggest a versatile mechanism for the regulation of actin assembly in FAs subjected to various combinations of biochemical and mechanical cues.

To migrate efficiently in different tissues, cells must sense and adapt to variations in the mechanical properties of their environment. In this adaptive process, focal adhesions (FAs) can strengthen their link with the extracellular matrix (ECM) and the actomyosin stress fibers[1–3]. FAs are composed of transmembrane integrins that mechanically couple the ECM to the actomyosin cytoskeleton, via a variety of actin binding proteins (ABPs)[2,4–6].

The mechanical coupling of stress fibers to FAs is highly regulated. The anchoring of actin filaments to FAs can be modulated by the degree of engagement of a molecular clutch composed of sliding layers of interacting proteins, including ABPs[1,7]. The regulation of the polymerization of the actin filaments that compose the stress fibers may also determine the level of their mechanical coupling to FAs[2]. Interestingly, FAs associated with elongating dorsal stress fibers are associated with low traction forces, whereas FAs associated with slowly elongating ventral stress fibers are associated with high traction forces[8]. This inverse correlation between the elongation of the actin filaments and the transmission of force to the ECM sheds light on the

importance of force-dependent ABPs which control actin assembly in FAs.

Biochemical and cellular studies have described a variety of ABPs associated with FAs that regulate the elongation of actin filament barbed ends. Early in vitro studies showed that the vasodilator-stimulated phosphoprotein (VASP) nucleates actin filaments and assembles them into bundles[9]. More recent studies demonstrate that VASP can also elongate actin filament barbed ends in a processive manner[10–12]. Similarly, a series of studies suggested that formins are involved in the processive elongation of stress fibers in cells[13,14]. Talin and vinculin, which associate in response to the actomyosin force, could link force sensing to the control of actin polymerization in FAs.

Vinculin is an autoinhibited ABP in which the single actin-binding domain (ABD), called vinculin tail ($V_t$), is masked by an intramolecular interaction with vinculin head ($V_h$)[15]. Biochemical studies showed that isolated $V_t$ binds actin filaments and assembles them into bundles[16,17]. We showed that $V_t$ also caps

[1]Université Paris-Saclay, CEA, CNRS, Institute for Integrative Biology of the Cell (I2BC), Gif-sur-Yvette, France. [2]Institute of Experimental and Clinical Pharmacology and Toxicology, Medical Faculty, University of Freiburg, Freiburg, Germany. [3]Centre for Integrative Biological Signalling Studies-CIBSS, University of Freiburg, Freiburg, Germany. [4]These authors contributed equally: Rayan Said, Clémence Nguyen-Vigouroux. ✉e-mail: christophe.leclainche@i2bc.paris-saclay.fr

actin filament barbed ends and nucleates actin filaments[18]. All these activities of $V_t$ are masked by $V_h$ in the autoinhibited conformation of vinculin.

Talin is a large ABP composed of a FERM domain, subdivided into F0, F1, F2, F3, and a rod domain made of 13 helical bundles (R1 to R13)[2]. The two major intramolecular interactions F3-R9 and F2-R12 keep talin

in an autoinhibited form[19,20]. Talin contains three ABDs. ABD1, ABD2 and ABD3 correspond to the head (F2F3), the central region (R4-R8) and the last C-terminal bundle (R13) respectively[21]. We previously demonstrated that the N-terminal ABD1 of talin blocks the elongation of actin filament barbed ends in low ionic strength conditions, whereas ABD2 and ABD3 do not affect actin dynamics[22]. Full-length talin is

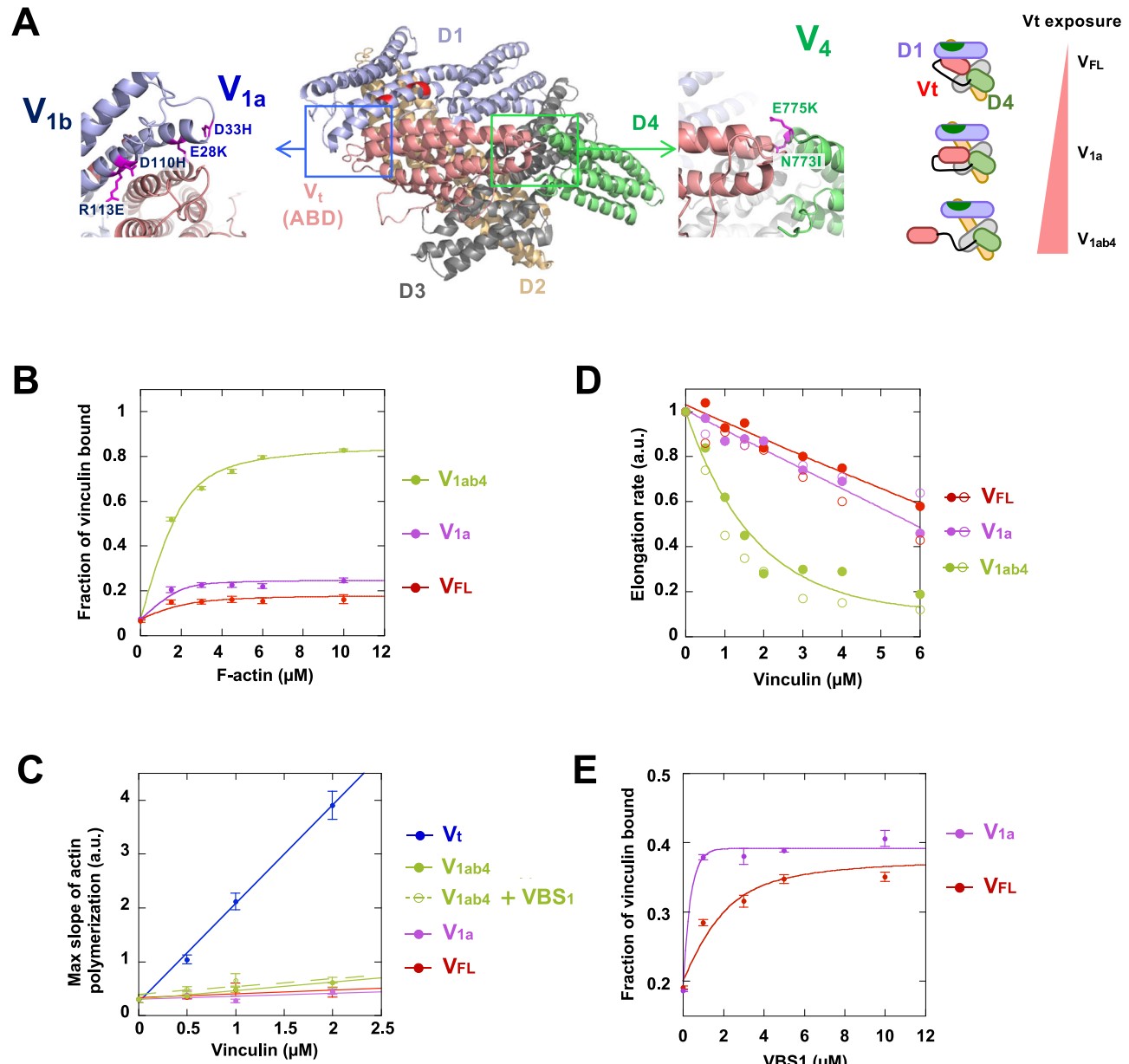

**Fig. 1 | The release of the autoinhibitory contacts of vinculin allows F-actin binding and barbed-end capping but not nucleation. A** Structure of vinculin featuring the subdomains and the auto-inhibitory contacts D1-$V_t$ (blue box) and D4-$V_t$ (green box). In D1, the double mutations E28K/D33H and D110H/R113E are referred to as $V_{1a}$ and $V_{1b}$ respectively. The double mutation N773I/E775K in D4 is referred to as $V_4$. The mutated amino acids are in magenta. A schematic representation of the vinculin mutants used in this study is shown on the right. We used the existing structure PDB:1TR2 (https://www.rcsb.org/structure/1tr2). **B** Quantification of the co-sedimentation of the indicated vinculin mutants (2 μM) in the presence of increasing concentration of F-actin. The fraction of vinculin bound in the pellet is plotted against F-actin concentration (Supplementary Fig. 1). **C** The maximal rate of spontaneous actin polymerization (1.5 μM, 10% pyrenyl-labeled) is plotted against the concentration of the indicated vinculin mutants (Supplementary Fig. 2). $V_t$ is used as a positive control for nucleation. **D** Actin filament barbed

end elongation is measured in the presence of spectrin-actin seeds (100 pM), 2 μM actin (10% pyrenyl-labeled), and increasing concentrations of the indicated vinculin mutants (Supplementary Fig. 3). The fraction of barbed end elongation is the ratio between the elongation rates in the presence and absence of vinculin. Open and closed circles indicate two independent experiments in the same conditions. **C**, **D** Actin polymerization and elongation rates are expressed in arbitrary units (a.u.). **E** The binding of the indicated vinculin constructs (2 μM) to F-actin (10 μM) is measured in a co-sedimentation assay in the presence of increasing concentrations of talin VBS$_1$ (Supplementary Fig. 5). The fraction of vinculin bound to F-actin is plotted against the concentration of VBS$_1$. In (**B**–**E**), data are mean ± SE, three independent experiments. The curves are not fits of the experimental points by a model, but are drawn manually to reflect the trend of the data, in order to make the data easier to read. Source data are provided as a Source Data file.

inactive because ABD1 is inhibited by the F3-R9 intramolecular interaction.

Biochemical and structural studies revealed the presence of 11 vinculin binding sites (VBSs) buried in some of the 13 helical bundles of talin[23]. The stretching of single molecules of talin revealed that force exposes several of these 11 cryptic VBSs along talin rod domain[24–26]. Using an in vitro reconstitution approach, we demonstrated that the actomyosin force is sufficient to stretch talin, allowing its binding to vinculin[27]. The release of talin autoinhibition also favors the low-affinity constitutive binding of vinculin[28].

In order for the $V_h$ domain of vinculin to interact with talin VBSs, it is necessary to release it from its autoinhibitory interaction with $V_t$[29]. Although the mechanism of activation of vinculin is not fully understood, several studies show that once vinculin is bound to talin, the interaction of $V_t$ with actomyosin keeps vinculin under tension in its open conformation[30–34]. In this mechanism, the actomyosin-dependent talin-vinculin complex acts as a catch-bond that releases $V_t$ to strengthen the anchoring to actomyosin whose contraction initially induced its formation[27,34,35].

Because force is required to trigger vinculin association to talin, and because the two proteins are autoinhibited, it has so far been impossible to determine the ability of the talin-vinculin complex to regulate actin polymerization.

Here, we use a series of talin and vinculin mutants that associate constitutively into a stable complex. By combining kinetic studies in fluorescence spectroscopy, actin binding assays, single actin filament observations in TIRF microscopy and FRAP experiments in cells, we show how these mutants and their complexes interact with actin filaments and regulate their polymerization in vitro, and control the elongation of stress fibers in cells. Altogether our data reveal a force-dependent mechanism of actin assembly in FAs.

## Results

### The release of the two autoinhibitory contacts of vinculin allows F-actin binding and barbed-end capping but not nucleation

Before starting the study of the talin-vinculin complex, we deciphered the complex mechanism that links the autoinhibition of vinculin and its activities. Indeed, biochemical and structural studies have revealed that the auto-inhibition of vinculin is controlled by two interfaces between $V_t$ and the D1 and D4 subdomains of the head[18,36,37] (Fig. 1A).

In order to disrupt specifically both autoinhibitory interfaces, we designed a series of point mutations of amino acids in D1 and D4 that interact with $V_t$. Because the contact surface between $V_t$ and D1 is significantly larger than the D4-$V_t$ one, we mutated two separate groups of charged amino acids in the D1-$V_t$ interface: E28K, D33H referred to as $V_{1a}$ and D110H, R113E referred to as $V_{1b}$ (Fig. 1A). Their combination is referred to as $V_{1ab}$. In D4, we designed a double mutation N773I, E775K called $V_4$ (Fig. 1A). In most of our comparative studies, we used full-length vinculin ($V_{FL}$), the isolated ABD $V_t$ and the $V_{1a}$ and $V_{1ab4}$ mutants.

We first characterized these vinculin mutants by measuring their binding to actin filaments (F-actin) using a cosedimentation assay. As previously reported, the autoinhibited $V_{FL}$ has low affinity for F-actin[18] (Fig. 1B, Supplementary Fig. 1A). Among the mutants, $V_{1ab4}$ displays a high affinity for F-actin (Fig. 1B and Supplementary Fig. 1B), while $V_{1a}$ displays low to intermediate binding level (Fig. 1B and Supplementary Fig. 1C). Altogether, these data demonstrate that the combined release of D4 and D1 efficiently exposes $V_t$ (Supplementary Table 1).

We then tested the effect of these vinculin mutants on actin nucleation by measuring the acceleration of the kinetics of pyrenyl-labeled actin assembly in low ionic strength condition, which corresponds to 25 mM KCl, while the salt concentration normally used by the vast majority of studies is 50 mM KCl and sometimes 100 mM KCl. Several studies have revealed regulatory mechanisms and activities of FA proteins, such as vinculin, talin and VASP, by varying the ionic

strength of in vitro assays[9,18,20]. We first confirmed that $V_t$ nucleates actin filament in a dose-dependent manner[18] (Fig. 1C and Supplementary Fig. 2A). In contrast, $V_{FL}$, $V_{1a}$, and $V_{1ab4}$ do not show significant actin nucleation activities (Fig. 1C and Supplementary Fig. 2B-D, Supplementary Table 1). Although the activity is very weak, the kinetics showed that $V_{1ab4}$ displays a measurable nucleation activity (Supplementary Fig. 2D), suggesting that the release of $V_t$ from both D1 and D4 is necessary to nucleate actin, but not sufficient to fully expose this activity of $V_t$.

These two series of experiments lead to an apparent contradiction, since the release of the 2 contacts D1-$V_t$ and D4-$V_t$ is sufficient to expose $V_t$ and allow it to bind to F-actin (Fig. 1B), but not to nucleate actin filaments (Fig. 1C). We hypothesized that, in addition to releasing the D1-$V_t$ and D4-$V_t$ contacts, F-actin, which is present in cosedimentation assays, but absent at the beginning of the nucleation assays, contributes to the opening of vinculin by binding to $V_t$. To test this hypothesis, we compared the ability of vinculin mutants to cap pre-existing actin filament barbed ends. The effect of vinculin mutants on barbed end capping was first assessed by measuring the inhibition of pyrenyl-labeled actin polymerization in the presence of spectrin-actin seeds[18]. Although $V_{FL}$ and $V_{1a}$ display only weak activity (Fig. 1D, Supplementary Fig. 3A, B), $V_{1ab4}$ strongly inhibits the elongation of pre-existing actin filament barbed ends, in agreement with our hypothesis (Fig. 1D, Supplementary Fig. 3C, and Supplementary Table 1).

### An isolated vinculin-binding domain of talin promotes vinculin binding to F-actin but has no effect on nucleation

Before determining the activity of the talin-vinculin complex, it was necessary to determine whether and how the simple binding of talin to vinculin, independently of the activities of talin ABDs, influences the activities of vinculin. We therefore used a short domain of talin, called $VBS_1$ (talin 482-636) corresponding to R1 deleted from its last helix, which exposes one VBS[18] (Supplementary Fig. 4).

First, $VBS_1$ does not increase the very weak nucleation activity of $V_{1ab4}$, which remains negligible compared to that of $V_t$ (Fig. 1C and Supplementary Fig. 2A, E). This assay, in which F-actin is initially absent, further supports the importance of F-actin as a co-activator of vinculin.

We then tested the effect of $VBS_1$ (0–10 μM) on the ability of vinculin to bind to F-actin, using a cosedimentation assay (Fig. 1E and Supplementary Fig. 5). $VBS_1$ induces an increase in $V_{FL}$ binding to F-actin (Fig. 1E and Supplementary Fig. 5A). The weakening of D1-$V_t$ increases the effect of $VBS_1$, which results in a higher apparent affinity of $V_{1a}$ for F-actin (Fig. 1E and Supplementary Fig. 5B).

### Talin with exposed VBSs promotes vinculin-dependent barbed-end capping but has no effect on nucleation

After determining the contribution of an isolated talin VBS to the activity of vinculin, we designed a series of full-length talin constructs (TΔ1, TΔ2, TΔ3) in which VBSs are exposed (Supplementary Figs. 6A and 4), in order to study the activity of a constitutive talin-vinculin complex harboring the ABDs of both proteins. We first designed a construct, referred to as TΔ1, in which the helix 5 of the helical bundle R1 is deleted, which exposes the same VBS than that of the construct called $VBS_1$ used in Fig. 1. Talin TΔ2 has been designed to expose two consecutive VBSs in R3, thanks to the deletion of the 2 helices 10 and 13 located on both sides of the two VBSs. Talin TΔ3 combines the mutations of TΔ1 and TΔ2. We first verified the efficiency of deletions performed in TΔ1, TΔ2, and TΔ3 using a micropatterning-based binding assay adapted from our previous work[27,38,39]. Hence, TΔ1, TΔ2, TΔ3 constructs immobilized on micropatterned surfaces recruit the head of vinculin fused to EGFP ($V_h$-EGFP) more efficiently than the wild-type $T_{FL}$ (Supplementary Fig. 6B, C). Because it is impossible to distinguish the relative contribution of each protein in the talin-vinculin complex to F-actin

binding, we did not perform cosedimentation assays for this part of the study.

We then determined the contribution of talin to the barbed-end capping activity of the talin-vinculin complex. To this aim, we combined the talin mutants with the $V_{1a}$ mutant of vinculin which does not efficiently cap actin barbed ends alone (Fig. 1D). In the presence of a fixed concentration of $V_{1a}$, addition of increasing concentrations of T$\Delta$1, T$\Delta$2 and T$\Delta$3 induces a dose-dependent inhibition of barbed-end elongation in a spectrin-actin seed assay (Supplementary Fig. 6D, Supplementary Fig. 7B–D, and Supplementary Table 1). Wild-type full-length talin ($T_{FL}$) also induces barbed-end inhibition in the presence of $V_{1a}$, but with a lower efficiency than T$\Delta$1, T$\Delta$2 and T$\Delta$3 (Supplementary Fig. 6D, Supplementary Fig. 7A, and Supplementary Table 1). This effect is not due to talin constructs alone and requires the presence of $V_{1a}$ since $T_{FL}$, T$\Delta$1, T$\Delta$2, and T$\Delta$3 alone had no effect on barbed end elongation (Supplementary Figs. 6D and 8A–D). The fact that the fully autoinhibited $V_{FL}$ does not combine with $T_{FL}$, T$\Delta$1, T$\Delta$2, and T$\Delta$3 to inhibit barbed end elongation efficiently confirms that vinculin activation is required for this activity (Supplementary Fig. 9A–E).

We then tested the ability of the talin-vinculin complex to stimulate actin assembly by combining the talin mutants with the vinculin mutant $V_{1ab4}$. The mutant $V_{1ab4}$ appeared to be the ideal choice here because it is close to the fully open conformation (Fig. 1A). We hypothesized that the weak nucleation activity of $V_{1ab4}$, (Fig. 1C and Supplementary Fig. 2D), could be increased by talin binding to vinculin head and the activity of talin ABDs. However, kinetic assays containing $V_{1ab4}$ and $T_{FL}$, T$\Delta$1, T$\Delta$2, or T$\Delta$3, do not reveal a significant nucleation activity compared to the activity of $V_t$ alone, and it is not higher than that of $V_{1ab4}$ alone (Supplementary Figs. 6E and 10A–D, and Supplementary Table 1).

### A talin-vinculin complex, in which all autoinhibitory contacts are released, nucleates actin filaments transiently capped at their barbed ends

The results obtained with the previous talin mutants may only partially reflect the activity of the talin-vinculin complex because talin remains autoinhibited by the F3-R9 and F2-R12 contacts[19,20]. We previously demonstrated that the F3-R9 contact masks the capping activity of talin ABD1[22]. Therefore, we hypothesized that releasing these autoinhibitory contacts in talin may unravel additional activities of the talin-vinculin complex. To test this hypothesis, we designed three talin constructs, called T$\Delta$AI, T$\Delta$1$\Delta$AI and T$\Delta$1$\Delta$AI$\Delta$ABD2 (Fig. 2A and Supplementary Fig. 4). T$\Delta$AI corresponds to $T_{FL}$ deleted from the region encompassing the auto-inhibitory (AI) modules R9 and R12. T$\Delta$1$\Delta$AI combines the deletion of the AI region ($\Delta$AI) and the deletion that exposes a VBS in R1 ($\Delta$1). T$\Delta$1$\Delta$AI$\Delta$ABD2 combines the $\Delta$AI and $\Delta$1 deletions with the deletion of the second actin binding domain ($\Delta$ABD2).

In a spectrin-actin seed assay, increasing concentrations of T$\Delta$AI and T$\Delta$1$\Delta$AI stimulate the barbed-end capping activity of $V_{1a}$ but not that of $V_{FL}$ (Fig. 2B, Supplementary Fig. 11A–E, and Supplementary Table 1). Altogether these experiments show that releasing the autoinhibitions of talin favors its association with vinculin to form a complex that caps actin filament barbed ends, provided that vinculin autoinhibition is also weakened. The fact that T$\Delta$1$\Delta$AI is much more effective than T$\Delta$AI for this activity further indicates that the release of the autoinhibition of talin adds to the VBS exposure in R1 to promote vinculin binding. The strong barbed-end capping activity of the complex made of T$\Delta$1$\Delta$AI$\Delta$ABD2 and $V_{1a}$, but not $V_{FL}$, also indicates that talin ABD2 is not required for this activity (Fig. 2C, Supplementary Fig. 12A–C, and Supplementary Table 1). The number and position of exposed VBSs influence the barbed end capping activity of the complex since T$\Delta$2$\Delta$AI, in which two VBSs are exposed in R3, and T$\Delta$3$\Delta$AI, in which three VBSs are exposed in R1 and R3, combine efficiently with $V_{FL}$ to cap actin filaments (Supplementary Fig. 13A–E and

Supplementary Table 1), in contrast with T$\Delta$1$\Delta$AI, in which only one VBS is exposed in R1 (Fig. 2B).

To confirm that this inhibition of elongation is indeed due to an interaction with the barbed end, we compared the ability of the T$\Delta$1$\Delta$AI-$V_{1a}$ complex to inhibit the elongation of spectrin-actin seeds and the elongation mediated by the FH1-FH2 fragment of the formin mDia1, which is known to elongate filaments while residing processively at their growing barbed end, thus protecting the filaments from barbed-end capping. Our quantifications show that actin filaments polymerizing in the presence of mDia1 are less effectively inhibited by the T$\Delta$1$\Delta$AI-$V_{1a}$ complex than filaments with free barbed ends, confirming the binding of the complex to the barbed end (Supplementary Fig. 14A–C).

We then tested the effect of complexes composed of T$\Delta$AI, T$\Delta$1$\Delta$AI, and T$\Delta$1$\Delta$AI$\Delta$ABD2 and vinculin on actin nucleation. Increasing concentrations of T$\Delta$AI and T$\Delta$1$\Delta$AI stimulate actin assembly in the presence of $V_{1ab4}$ but not in the presence of $V_{FL}$ (Fig. 2D, Supplementary Fig. 15A–D, and Supplementary Table 1). As for the capping activity (Fig. 2B), T$\Delta$1$\Delta$AI is much more effective than T$\Delta$AI for the nucleation activity, indicating that, in addition to the release of talin autoinhibition, VBS exposure is required for talin and vinculin to form a nucleation complex (Fig. 2D and Supplementary Fig. 15C, D). The number and position of exposed VBSs influence this activity of the complex since a complex made of $V_{1ab4}$ and T$\Delta$3$\Delta$AI, in which three VBSs are exposed in R1 and R3, is slightly more active than complexes made of $V_{1ab4}$ and T$\Delta$1$\Delta$AI or T$\Delta$2$\Delta$AI, in which one or two VBSs are exposed respectively. This effect is visible at high concentration of T$\Delta$3$\Delta$AI (Supplementary Fig. 16A–F and Supplementary Table 1).

This nucleation activity is strong at 25 mM KCl and remains significant at 50 mM KCl but disappears at 75 mM KCl, indicating the electrostatic nature of interactions between talin, vinculin and actin (Supplementary Fig. 17A, B). The conditions we used to measure the stimulation of actin assembly by talin and vinculin are close to those generally used to study actin polymerization activities in vitro with purified proteins. However, these conditions (20 °C, pH = 7.8, 25 mM KCl) are not physiological. We therefore performed kinetic assays under physico-chemical conditions as close as possible to those found in mammalian cells, i.e. 37 °C, pH = 7, 100 mM KCl. Under these conditions, spontaneous actin polymerization is faster and the stimulation of actin polymerization by the combined action of T$\Delta$1$\Delta$AI or T$\Delta$3$\Delta$AI and $V_{1ab4}$ is strong (Supplementary Fig. 18A, B). Importantly, in cells, polymerizable actin is complexed with profilin, which prevents spontaneous nucleation of actin filaments and restricts the formation of new filaments at specific sites, generally associated with a membrane-bound structure stimulated by chemical or mechanical signals. We therefore tested the effect of profilin on nucleation activity. Although profilin does not completely abolish this activity, the very strong reduction observed suggests that an additional mechanism is involved in the dissociation of profilin (Supplementary Fig. 18B).

The fact that combinations of activated vinculin $V_{1ab4}$ with auto-inhibited talin (T$\Delta$1, T$\Delta$2, and T$\Delta$3) do not stimulate actin assembly indicates that autoinhibited ABDs in talin play a critical role in this mechanism. To determine the relative importance of the ABDs of talin and vinculin for the activity of the complex, we tested various constructs of the two proteins deleted for ABDs and we added competitors of ABDs to polymerization reactions. First, we showed that talin ABD2 is not necessary since T$\Delta$1$\Delta$AI$\Delta$ABD2, lacking ABD2, efficiently combines with $V_{1ab4}$ to stimulate actin assembly (Fig. 2E, Supplementary Fig. 19A–C, and Supplementary Table 1). We also compared the ability of talin constructs containing VBSs and either ABD1 or ABD3 to stimulate actin assembly in the presence of $V_{1ab4}$. We found that ABD1 and ABD3 can combine with vinculin to stimulate actin polymerization, suggesting redundancy between ABD1 and ABD3 (Supplementary Fig. 20A). Interestingly, the addition of the isolated R9 domain of talin

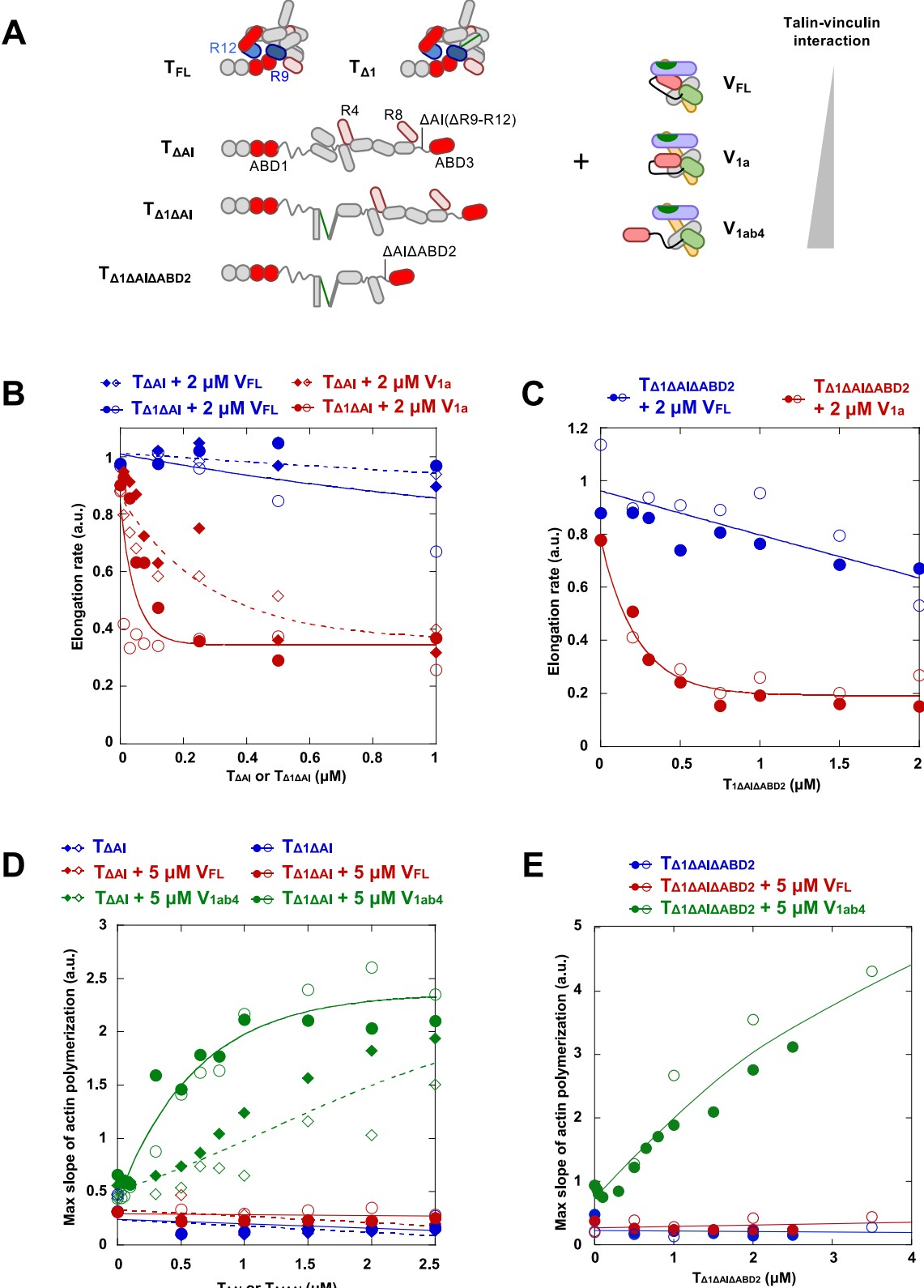

to a polymerization reaction containing TΔ1ΔAIΔABD2 and V1ab4 reduces the stimulation of actin assembly (Supplementary Fig. 20B). This result confirms that ABD1, which is masked by R9 in inactive talin[22], is involved in this activity. Finally, combinations of various activated talin constructs with vinculin head (Vh), which lacks its ABD Vt, do not block the elongation of actin filament barbed ends or stimulate actin assembly, indicating a critical role of Vt in all activities of

the talin-vinculin complex (Supplementary Fig. 21A, B). Altogether our observations indicate that ABD1 and ABD3 domains of talin, as well as the Vt domain of vinculin, are involved in the activities of the complex.

To confirm the mechanism by which the talin-vinculin complex regulates actin assembly, we observed in TIRF microscopy the actin filaments produced by the complex made of V1ab4 and TΔ1ΔAI or TΔ1ΔAIΔABD2 described above.

**Fig. 2 | A complex of talin and vinculin, in which all autoinhibitory contacts are released and VBSs are exposed, nucleates and caps actin filaments.**
(A) Schematic representation of the talin and vinculin constructs used in (B–E).
B, C Actin filament barbed end elongation, expressed in arbitrary units (a.u.), is measured in the presence of 100 pM spectrin-actin seeds, 1.5 μM actin (10% pyrenyl-labeled), 2 μM $V_{FL}$ (blue) or 2 μM $V_{1a}$ (red) and increasing concentrations of (B) $T_{\Delta AI}$ (dash line) or $T_{\Delta 1 \Delta AI}$ (solid line), or (C) $T_{\Delta 1 \Delta AI \Delta ABD2}$. The fraction of barbed end elongation was then calculated as the ratio between the elongation rate in the presence of the indicated combinations of vinculin and talin and the elongation rate of actin alone. **D**, **E** The maximal rate of spontaneous actin polymerization (2.5 μM, 10% pyrenyl-labeled), expressed in arbitrary units (a.u.), is plotted as a function of increasing concentrations of (**D**) $T_{\Delta AI}$ or $T_{\Delta 1 \Delta AI}$, (**E**) $T_{\Delta 1 \Delta AI \Delta ABD2}$ alone and in the presence of $V_{FL}$ or $V_{1ab4}$. In **B**–**E**, open and closed symbols indicate two independent experiments in the same conditions. The curves are not fits of the experimental points by a model, but are drawn manually to reflect the trend of the data, in order to make the data easier to read. Source data are provided as a Source Data file.

At high salt concentration (100 mM KCl), which restricts the activity of the talin-vinculin complex to the capping of the barbed ends only, the filaments show pauses interrupting the elongation of their barbed end (Fig. 3A, D and Supplementary Movie 1). Quantifying the distribution of these pause times enabled us to estimate a dissociation rate constant of the complex from the barbed end of 0.006 s$^{-1}$ ($t_{1/2}$ = 115.8 s) (Fig. 3E). Quantification of the elongation rate between pauses revealed a slight reduction in the presence of $V_{1ab4}$ and $T\Delta 1 \Delta AI$ reflecting the presence of remaining short capping events that escaped our analysis or an effect on the elongation other than barbed end capping (Fig. 3F). In low salt concentration, favoring actin nucleation by the talin-vinculin complex (Fig. 2D, E), representative images and quantifications showed that the $T\Delta 1 \Delta AI$-$V_{1ab4}$ complex induces the formation of a higher number of actin filaments than $T\Delta 1 \Delta AI$ alone or $V_{1ab4}$ alone (Fig. 3B, G and Supplementary Movie 2). We found the same result for $T\Delta 1 \Delta AI \Delta ABD2$-$V_{1ab4}$ complex (Fig. 3C, H and Supplementary Movie 3). The analysis of actin filament elongation showed that they undergo frequent pauses reflecting barbed end capping events. The exponential fit of the distribution of the pause duration gave an estimated dissociation rate constant of 0.0056 s$^{-1}$ ($t_{1/2}$ = 124.2 s) for $T\Delta 1 \Delta AI$-$V_{1ab4}$ and 0.007 s$^{-1}$ ($t_{1/2}$ = 99.6 s) for $T\Delta 1 \Delta AI \Delta ABD2$-$V_{1ab4}$ (Fig. 3E). Quantification of the elongation rate between pauses did not reveal any effect of $V_{1ab4}$ and $T\Delta 1 \Delta AI$ (Fig. 3F).

To explore the mechanisms underlying the above microscopy observations, we followed fluorescent molecules of vinculin, talin and actin in real time during the nucleation and capping of individual actin filaments. We first immobilized pre-formed complexes of $T\Delta 1 \Delta AI$-Alexa594 and $V_{1ab4}$-Alexa647 at low concentration on a glass surface passivated with PLL-PEG and added actin-Alexa488 under conditions that allowed nucleation and capping. This experimental setup allowed us to capture sequences of events in which spots, containing both talin and vinculin, associate with actin and release a growing actin filament (Fig. 4A, Supplementary Movie 4). In this experiment, as in FAs that form in cells, talin, and vinculin are dimers that assemble with high stoichiometry. The observed spots therefore contain several molecules of $T\Delta 1 \Delta AI$ and $V_{1ab4}$ that assemble via the exposed VBS of $T\Delta 1 \Delta AI$ but also via other VBSs with lower affinity as indicated by the activity of the $T\Delta AI$-$V_{1ab4}$ experiments (Fig. 2D). The stationary fiduciary marks along actin filaments indicate that the talin-vinculin complex remains associated to the filament near the pointed end while the barbed end is free to grow on the other side (black arrow head, Fig. 4A). Interestingly, nucleation events are often characterized by the existence of a delay between actin recruitment by a talin-vinculin complex and the elongation of a filament (Fig. 4B and Supplementary Movie 5). In this series of experiments, pre-existing filaments are also captured and capped at their barbed end (Fig. 4C and Supplementary Movie 6). Barbed-end capping is transient, as many of the capped filaments are eventually left free to elongate after a delay (Fig. 4D and Supplementary Movie 7). These activities lead to a distribution of talin-vinculin complexes near the pointed end of the filaments, following their nucleation, at the barbed end of the filaments, following their capping, or on the side of the filaments (Fig. 4E). Quantification of the delay between actin recruitment and elongation allows to determine a rate (0.0047 s$^{-1}$, $t_{1/2}$ = 147 s, Fig. 4E), that is close to the dissociation rate of the barbed ends from the talin-vinculin complex observed previously (Fig. 3E),

suggesting that the talin-vinculin complex nucleates filaments initially capped at their barbed end.

## The association of talin and vinculin contributes to the assembly of stress fibers in cells

To determine the effect of talin-vinculin complex formation on actin assembly in cells, we compared Hela cells co-expressing fluorescent wild-type talin and vinculin (BFP-Talin$_{FL}$ and mCherry-Vinculin$_{FL}$) with cells co-expressing BFP-$T\Delta 1 \Delta AI$ and mCherry-$V_{1ab4}$ mutants. We first quantified FA area labeled with mCherry-vinculin and the width of stress fibers labeled with Alexa488-phalloidin. Cells expressing $T\Delta 1 \Delta AI$ and $V_{1ab4}$ have larger FAs associated with wider stress fibers (Supplementary Fig. 22B, C, D), compared to cells expressing wild-type talin and vinculin (Supplementary Fig. 22A, C, D).

To determine whether the constitutive formation of the talin-vinculin complex enhances stress fiber assembly dynamics, we compared the rate at which EGFP-actin-containing stress fibers elongate by FRAP in Hela cells co-expressing wild-type BFP-talin and mCherry-vinculin, and in Hela cells co-expressing BFP-$T\Delta 1 \Delta AI$ and mCherry-$V_{1ab4}$ mutants (Fig. 5A, B and Supplementary Movie 8 and 9). As FAs slide retrogradely, we took this parameter into account and measured the rate of elongation of the stress fibers using the trailing edge of the FAs as a reference, revealed here by the fluorescence of mCherry-vinculin (Fig. 5C). Quantification of the relative actin-EGFP fluorescence recovery rate from the FAs indicates that stress fibers elongate faster in cells expressing $T\Delta 1 \Delta AI$ and $V_{1ab4}$ compared with cells expressing wild-type talin and vinculin (Fig. 5D). Taken together, our data indicate that binding of vinculin to talin in FAs leads to the stimulation of actin assembly.

## Discussion

The assembly of the talin-vinculin complex is a key step in the reinforcement of cell-matrix adhesions subjected to a mechanical stimulus such as the force generated by the actomyosin stress fibers. This complex strengthens the link with the tensile actomyosin cytoskeleton to resist the force it applies. However, the mechanism by which the force-dependent complex acts on actin filaments was poorly understood.

The mechanistic study of the talin-vinculin complex has long been limited by the fact that the assembly of the complex depends on the release of multiple autoinhibitory interactions, both in talin and in vinculin. Although it is established that force exposes cryptic vinculin binding sites in talin by stretching helical bundles, the mechanisms allowing the release of the additional autoinhibitory interactions between distant domains in talin are not fully understood. The structure of full-length talin in its autoinhibited conformation has been reported and revealed that, in addition to the known interaction between F3 and R9, the F2 domain interacts with the R12 domain[19,20]. These autoinhibitory contacts mask the binding interface of ABD1, corresponding to F2 and F3, with F-actin, while ABD2, composed of R4 and R8, is masked by an interaction with R3[20,22,28] (Fig. 6). The mechanism by which these contacts are disrupted is unclear. The binding of RIAM and PIP2 to talin F3 is known to disrupt the F3-R9 autoinhibition[40]. Once these two interdomain interactions are disrupted, talin can be pulled by actomyosin force, which stretches helical

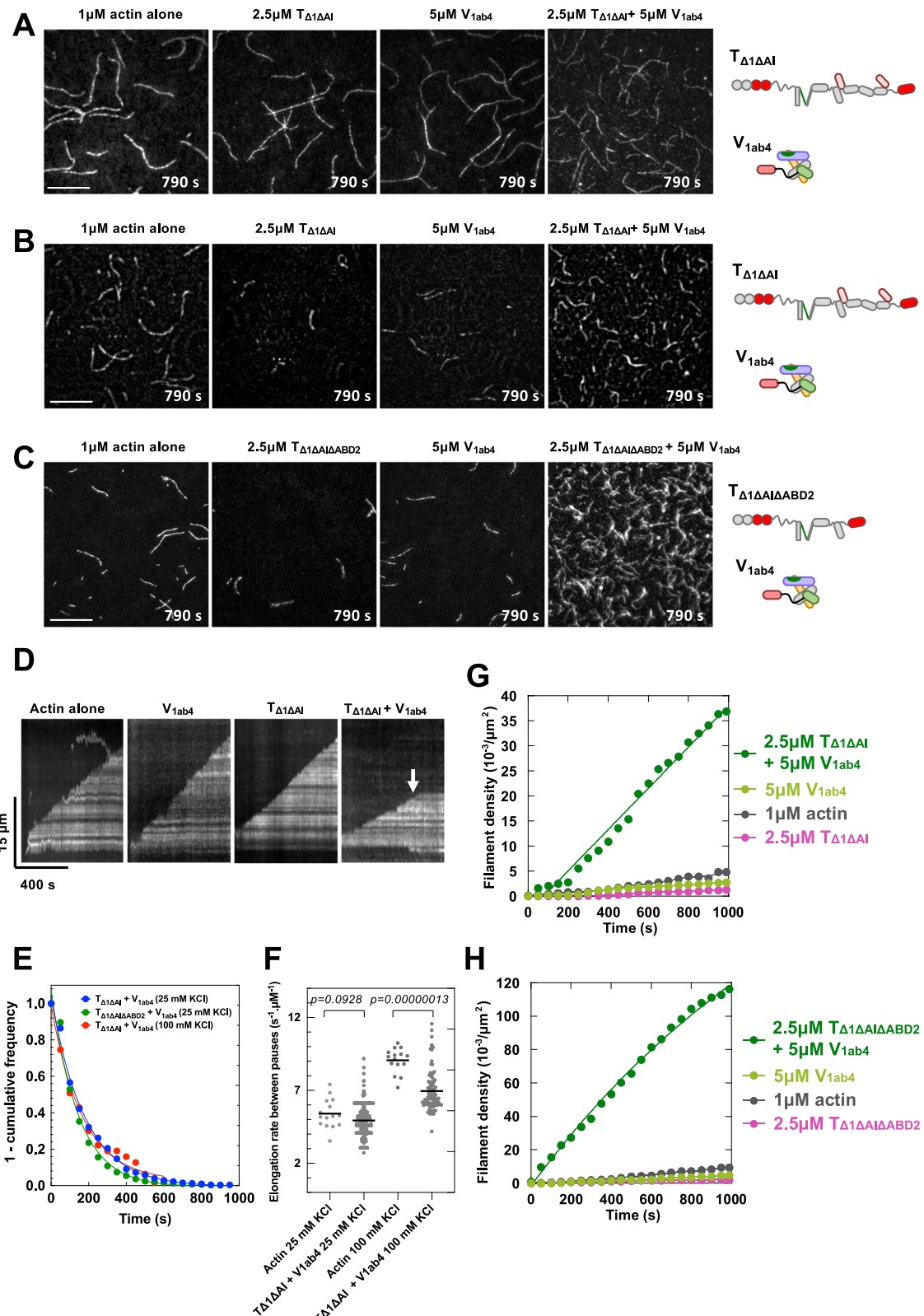

bundles to expose cryptic VBSs[24,25] (Fig. 6). Force application to the ABD3/R13 domain of talin could also contribute to disrupting its auto-inhibitions, but no experimental evidence of such a mechanism has been reported to date. This mechanism is possible because the ABD3 domain is exposed, at least partially, in the inactive talin, as suggested by the flexibility of the neighboring domains[20]. In this study, we show that combining various deletions, including the part of talin that

contains the R9 and R12 autoinhibitory domains, with the deletion of individual helices, which exposes VBSs, mimics the open and mechanically stretched conformation of talin (Fig. 6). Our results show that the release of talin autoinhibitions is sufficient to allow the binding of vinculin and form a complex that caps and nucleates actin, which explains the observation made by others that a mutation in talin that disrupts the F3-R9 interaction is sufficient to recruit vinculin in cells[28].

**Fig. 3 | Direct observation of the nucleation and capping of single actin filaments by the talin-vinculin complex. A–C** Single actin filaments observed in TIRF microscopy in the presence of 1 μM actin (5% Alexa488-labeled) alone and supplemented with 2.5 μM $T_{\Delta1\Delta AI}$ alone, 5 μM $V_{1ab4}$ alone and 2.5 μM $T_{\Delta1\Delta AI}$ with 5 μM $V_{1ab4}$ at 100 mM KCl (**A**) or 25 mM KCl (**B**), and in the presence of 1 μM actin (5% Alexa488-labeled) alone and supplemented with 2.5 μM $T_{\Delta1\Delta AI\Delta ABD2}$ alone, 5 μM $V_{1ab4}$ alone and 2.5 μM $T_{\Delta1\Delta AI\Delta ABD2}$ with 5 μM $V_{1ab4}$ at 25 mM KCl (**C**). Scale bar = 15 μm. See Supplementary Movie 1 (**A**), Supplementary Movie 2 (**B**), and Supplementary Movie 3 (**C**). Experiments in (**A**–**C**) were repeated twice independently with the same results. **D** Kymographs of single actin filaments in the presence of the indicated proteins from the experiment in (**A**). The arrow indicates a barbed end capping event. **E** Exponential fit of 1-cumulative frequency of barbed end pause times measured in the presence of 1 μM actin, 2.5 μM $T_{\Delta1\Delta AI}$ with 5 μM $V_{1ab4}$ (red line at 100 mM KCl, $t_{1/2} = 115.8 \pm 22$ s, $n = 349$, blue line at 25 mM KCl, $t_{1/2} = 124.2 \pm 10.8$ s, $n = 1023$) and 2.5 μM $T_{\Delta1\Delta AI\Delta ABD2}$ with 5 μM $V_{1ab4}$ (green line, $t_{1/2} = 99.6 \pm 13.5$ s, $n = 1619$) from experiments in (**A**–**C**). **F** Quantification of the elongation rates of single filaments between pauses in the experiments shown in (**A**), $n = 15$ for actin alone, $n = 263$ for $T_{\Delta1\Delta AI} + V_{1ab4}$ and (**B**), $n = 15$ for actin alone, $n = 91$ for $T_{\Delta1\Delta AI} + V_{1ab4}$. Data were analyzed using a two-sided unpaired t-test. **G**, **H** Quantification of filament density as a function of time from the experiments in (**B**) and (**C**). **A** is not quantified as nucleation is weak at 100 mM KCl. Source data are provided as a Source Data file.

Our data also support the findings that the association of talin and vinculin without tension is required for efficient nascent adhesion maturation[41]. This interaction likely corresponds to a low degree of saturation of the partially exposed VBSs in this open talin structure that has become very flexible. Indeed, exposure of a single VBS motif in R1 significantly increases vinculin binding as evidenced by the enhanced activity of the talin-vinculin complex.

Vinculin is autoinhibited by an intramolecular interaction between the head ($V_h$) and the tail ($V_t$) domains. Talin binding to the D1 subdomain of $V_h$ is controlled by the D1-$V_t$ interaction[29], whereas F-actin binding to $V_t$ is controlled by a more complex mechanism involving interactions of D1 and D4 with both sides of $V_t$[15,18,37]. The identification of several intermediate open conformations of vinculin suggests that the level of activation of vinculin is controlled by the number of auto-inhibitory contacts removed[42]. Our vinculin mutagenesis strategy confirmed that $V_t$ activities are controlled by different combinations of $V_h$-$V_t$ autoinhibitory contacts. We show that the disruption of D1-$V_t$ is not sufficient to expose the capping activity of $V_t$, which requires the combined disruption of D1 and D4 contacts with $V_t$, as previously suggested[18]. Early studies showed that the coincidental binding of talin and F-actin to D1 and $V_t$ respectively keep vinculin in its open conformation[31]. In agreement with this model, full-length vinculin dissociates rapidly from FAs after actomyosin inhibition, whereas $V_h$ remains stably associated, demonstrating that tensile actomyosin stress fibers bound to $V_t$ maintain the open conformation of talin-bound vinculin[32,33]. This observation is further explained by the discovery that $V_t$ forms a catch bond with F-actin in response to force[43]. It is therefore not surprising that the release of D1 and D4 from $V_t$ in our experiments releases the capping activity of vinculin that occurs in the presence of actin filaments, but results in very little actin nucleation initiated without actin filaments.

Solving the problem of releasing the auto-inhibitory contacts of talin and vinculin was a prerequisite for determining the activity that results from the combination of their ABDs. It is interesting to remember that the combination of ABPs within transient complexes is often found in important signal-responsive mechanisms for regulating actin dynamics. However, the results of such combinations cannot be predicted from the activities of the isolated proteins, as exemplified by the formation of branched actin filaments resulting from the combination of WASP-related proteins and Arp2/3[4]. Here, we show that the combination of talin ABDs with vinculin $V_t$ in a complex creates a machinery that nucleates actin filaments transiently capped at their barbed end (Fig. 6A). In this mechanism, talin and vinculin cooperate to form a new filament through a mechanism that is not yet understood but which probably involves the stabilization of actin nuclei, that form spontaneously in solution, by the ABDs of talin and vinculin. This mechanism leads to the transient barbed-end capping of the newly formed actin filaments. After being released from capping, the actin filament elongates by its free barbed end, while the complex remains attached to the side of the filament, near the pointed end (Fig. 6). It would be tempting to attribute to vinculin alone the nucleation and capping activities of the talin-vinculin complex, since $V_t$ has these activities[18,44]. The fact that a complex, composed of active vinculin and talin with exposed VBSs but masked ABD1, does not nucleate, shows unambiguously the importance of ABD1 for nucleation. Although the function of the ABD1 domain of talin is generally restricted to integrin activation in a membrane-bound conformation that does not allow its binding to actin, it is important to recall here that talin is in equilibrium between actin and the plasma membrane as evidenced by its retrograde movement in FAs[7]. ABD2 is not involved in capping or nucleation mechanisms as demonstrated by the strong activities of a mutant that does not contain this domain. However, ABD3 can also combine with $V_t$ to nucleate, suggesting a possible redundancy of ABD1 and ABD3 in this mechanism. It is interesting to note that the three ABDs of talin are redundant for the bundling of actin filaments[45], but are involved more selectively in the actin polymerization activities of talin in combination with vinculin. It is very likely that actin filaments nucleated by talin and vinculin will end up forming large bundles, but our microscopy experiments, where the filaments are short and spaced out, disadvantages this activity and we have not explored this direction already widely documented by recent studies[45,46].

The ability of a newly discovered nucleation machinery to form actin filaments in the presence of profilin has always been the subject of debate. Profilin prevents spontaneous polymerization of actin filaments in the cytoplasm where they would not play a role in producing force against the plasma membrane[4]. The generally accepted idea is that nucleation systems, located at the membrane and under the control of specific signaling pathways, trigger the polymerization of actin complexed with profilin. However, few nucleators are capable of nucleating actin filaments using profilin-actin complexes alone. Thus, the nucleation of branched filaments by the Arp2/3 complex or the nucleation of linear filaments by formins are severely inhibited in the presence of high profilin concentrations[47,48], as observed for the talin-vinculin complex. It is likely that a pool of free actin, resulting from the constant depolymerization of actin networks, is transiently available to feed nucleators, such as the talin-vinculin machinery that we described, before it associates to profilin. In the cell, the concentration of profilin is lower than that of actin[49]. Actin is therefore partially free, or bound to other proteins, such as thymosin ß4, which sequesters actin monomers, or to other proteins that could be compatible with nucleators.

In the context of a stress fiber anchored to a focal adhesion containing talin and vinculin, the release of actin filament barbed ends by talin-vinculin complexes, occurring with a half-time around 100 s, may contribute to the slow elongation of a stress fiber. This release could also simply reinforce stress fibers by feeding them with new actin filaments. It would be interesting to determine how the talin-vinculin complex cooperates with other ABPs in FAs, such as VASP, parvin, or tensin, to regulate actin assembly. The activities we have identified should generate filaments of different orientations in FAs. Actin filaments should be oriented with their barbed end facing the focal adhesion if the talin-vinculin complex caps their barbed end, or oriented with the barbed end towards the cell body

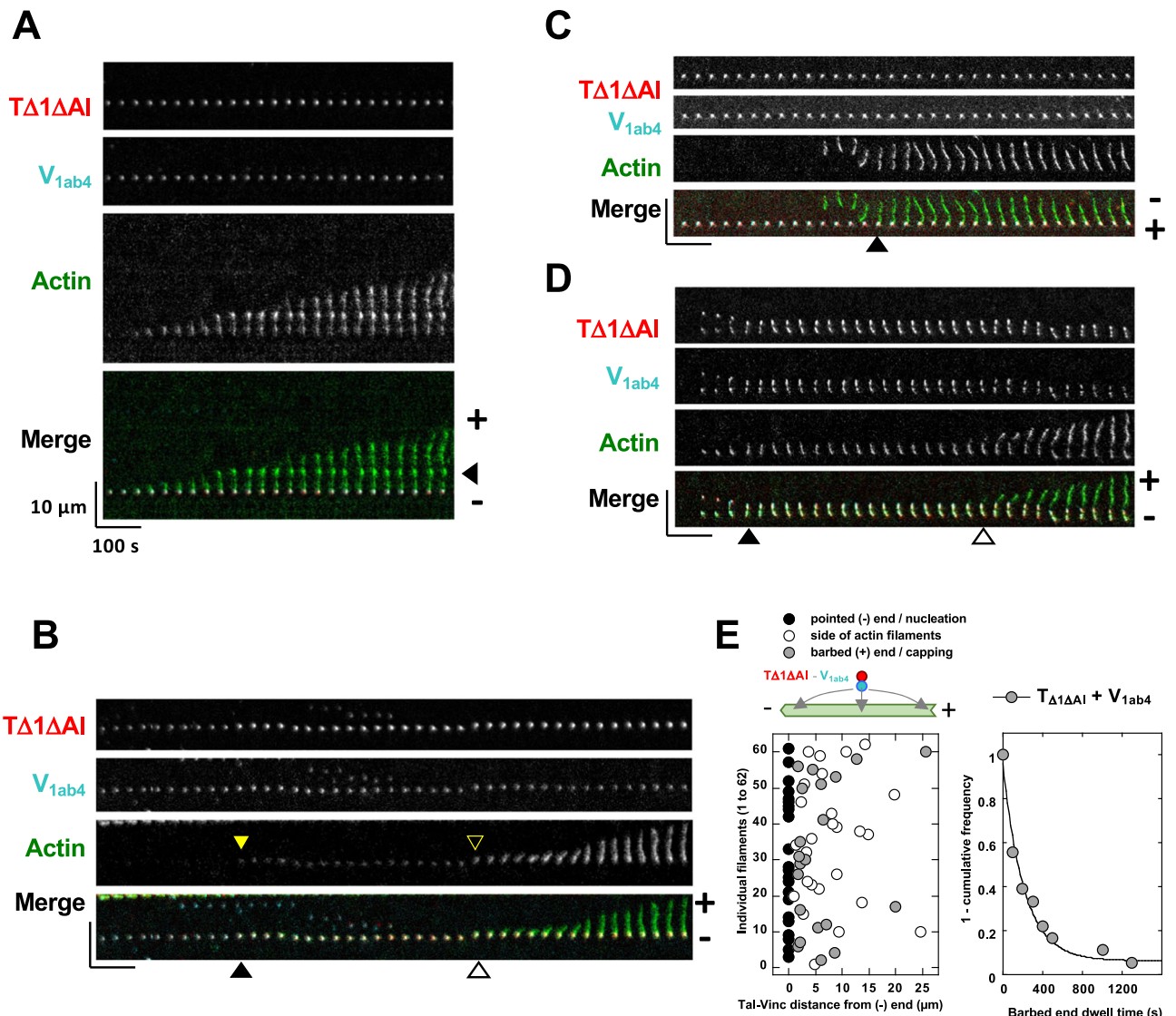

**Fig. 4 | Direct observation of talin, vinculin and actin during the nucleation and barbed-end capping of single actin filaments. A–E** 10 nM TΔ1ΔAI (78% Alexa594-labeled) and 50 nM $V_{1ab4}$ (18% Alexa647-labeled) are immobilized on a glass surface passivated with PLL-PEG, followed by the addition of 0.8 μM actin (5% Alexa488-labeled). **A** Kymographs showing the nucleation of an actin filament by a talin-vinculin complex. The stationary fiduciary mark indicated by the arrowhead on the right side indicate that the talin-vinculin complex remains associated near the pointed end (−) while the barbed end (+) grows on the other side (Supplementary Movie 4). **B** Recruitment of actin by a talin-vinculin complex followed by the elongation of a filament after a delay indicated by the arrowheads (Supplementary Movie 5). The delays are quantified in (**E**). **C** Kymographs showing the capture and barbed-end capping of an existing filament by a talin-vinculin complex (Supplementary Movie 6). **D** Kymographs showing the transient barbed-end capping of a filament by a talin-vinculin complex followed by its release (Supplementary Movie 7). Experiments in (**A–D**) were repeated twice independently with the same results. **E** (Left) Position of talin-vinculin complexes measured along 62 single filaments aligned on their pointed end (−). This quantification reveals that talin-vinculin complexes are found near the pointed end of newly nucleated filaments (black dots, $n = 22$), along the side of actin filaments (white dots, $n = 26$) or near the barbed end of capped filaments (gray dots, $n = 21$). Some of these events are on the same filaments. (Right) Exponential fit of 1-cumulative frequency of the time between actin recruitment by talin-vinculin complexes and barbed end elongation measured in the conditions described above and illustrated in (**B**), $t_{1/2} = 146.69 \pm 18.95$ s, $n = 18$. Source data are provided as a Source Data file.

or the leading edge if they have been nucleated and released. Although early electron microscopy studies concluded that the actin filaments making up the ends of stress fibers near the cell front are predominantly oriented with their barbed ends towards the membrane[50], later studies benefiting from the precision of cryoelectron tomography have somehow revised these conclusions. These studies show that the stress fibers in contact with the front part of a focal adhesion contain filaments whose barbed ends are oriented towards the leading edge of the cell, and filaments of mixed polarities at the rear of the adhesion[51], which is compatible with our findings (Fig. 6B). The same orientations have been observed in platelet pseudopodia[52]. The ability of the talin-vinculin complex to

generate actin structures containing antiparallel filaments should promote contractility by myosin II.

## Methods
### Recombinant cDNA constructs
cDNAs encoding for vinculin 1-1066 ($V_{FL}$), vinculin E28K/D33H ($V_{1a}$), vinculin E28K/D33H/D110H/R113E/N773I/E775K ($V_{1ab4}$) were synthesized and subcloned into the NcoI site of pET-3d by Genscript. pGEX-6P-1-$T_{F2F3R1R2R3}$ was obtained by PCR amplification of the talin-1 cDNA with primers Ta-196-Bam-CLC (Supplementary Table 2) and 3-primer-911-EcorI (Supplementary Table 2) and cloning into the BamH1 and EcoRI sites of pGEX-6P-1 plasmid (Cytiva). The pETM $T_{R2R3ABD3}$ plasmid

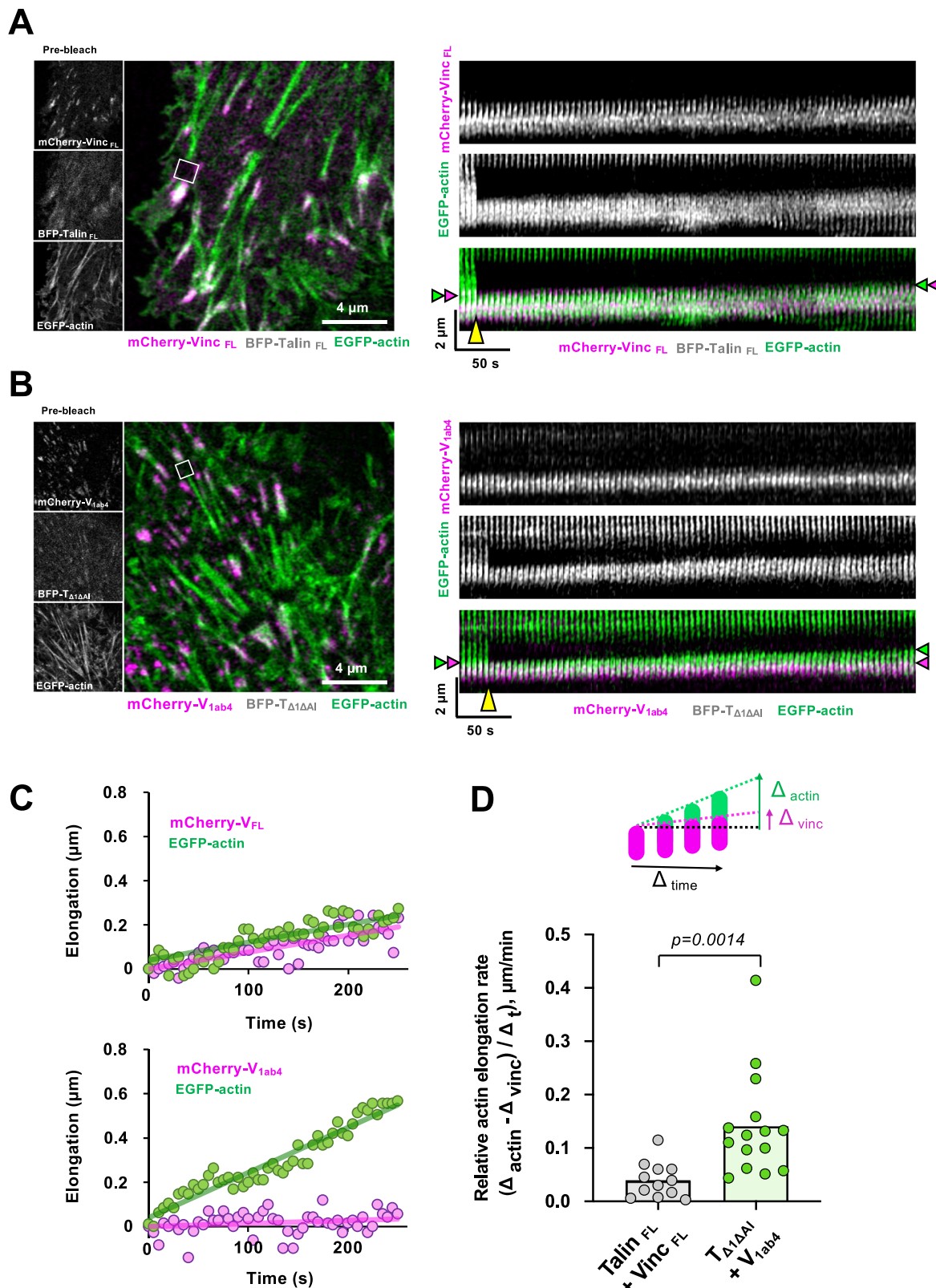

was obtained by PCR amplification of the talin-1 R2R3 cDNA with the o217 and o218 primers (Supplementary Table 2), and the talin-1 R13 (ABD3) cDNA with the o220 and o221 primers (Supplementary Table 2), followed by BamHI ligation and cloning into the KpnI and EcoRI sites of a pETM-11 plasmid (Novagen) previously modified to include a cDNA encoding a N-terminal Strep-tagII followed by Pre-Scission site (pETM-11sp). The pETM T$\Delta$1$\Delta$AI$\Delta$ABD2 plasmid was

obtained by PCR amplification of two talin-1 fragments obtained with the o174 and o224 primers (Supplementary Table 2), and the o225 and o228 primers (Supplementary Table 2), respectively, followed by their SmaI ligation and cloning into the NheI and NcoI sites of the pETM-11sp. Plasmids encoding for vinculin 879-1066 ($V_t$), talin-1 482–636 ($VBS_1$), talin-1 1655–1822 (R9), full-length talin-1 ($T_{FL}$), EGFP-Vh were described previously[18,22,38]. cDNAs of T$\Delta$AI, T$\Delta$1, T$\Delta$2, T$\Delta$3, T$\Delta$1$\Delta$AI,

**Fig. 5 | Constitutively active talin-vinculin complex accelerates stress fiber growth in Hela cells. A, B** Representative images of Hela cells expressing EGFP-actin together with mCherry-Vinculin$_{FL}$/BFP-Talin$_{FL}$ (**A**) or mCherry-V$_{1ab4}$/BFP-T$_{\Delta1\Delta AI}$ (**B**). Expression of mCherrry-vinculin, BFP-talin and EGFP-actin before bleaching is shown on the smaller left panels. Expression of mCherry-vinculin and EGFP-actin together with the bleached area (white square) is shown in the larger left panels. The corresponding kymographs of bleached stress fibers are shown on the right. Note that BFP-talin, mCherry-vinculin and EGFP-actin have been imaged before bleaching and only mCherry-vinculin and EGFP-actin have been imaged after bleaching to reduce time intervals. **C** The apparent elongation of EGFP-actin and retrograde movement of m-Cherry-vinculin are plotted as a function of time in the conditions mentioned in (**A**) and (**B**). **D** Scatter plot showing the quantification of the elongation rate of actin from the edge of vinculin-labeled FA in the conditions mentioned in (**A**) and (**B**). FRAP experiments were performed on 6 cells expressing mCherry-vinculin$_{FL}$ + BFP-talin$_{FL}$ + EGFP-actin, $n = 12$ measurements (2, 2, 2, 2, 2, 2) from 5 independent experiments performed in 5 different days, and 5 cells expressing mCherry-V$_{1ab4}$ + BFP-T$_{\Delta1\Delta AI}$ + EGFP-actin, $n = 15$ measurements (5, 1, 1, 3, 5) from 4 independent experiments performed in 4 different days. Each measurement represents the elongation rate of a stress fiber subtracted from the rate of retrograde sliding of the associated FA. Data were analyzed using a two-sided unpaired t-test. Source data are provided as a Source Data file.

T$\Delta2\Delta AI$, T$\Delta3\Delta AI$ were constructed and subcloned into pET-29a(+) by Genscript. The constructs were verified by sequencing. See Supplementary Fig. 4 for the detail of the talin constructs.

The pBFP-Talin$_{FL}$ and pBFP-T$_{\Delta1\Delta AI}$ plasmids were obtained through Gibson assembly of the pTagBFP-C1 backbone amplified by PCR with the primers Backbone-TagBFP-fwd and Backbone-TagBFP-rev (Supplementary Table 2), and the T$_{FL}$ and T$_{\Delta1\Delta AI}$ inserts amplified with the primers Insert-Talin-fwd and Insert-Talin-rev (Supplementary Table 2), using T$_{FL}$ and T$_{\Delta1\Delta AI}$ cDNAs as templates respectively. The pmCherry-Vinculin$_{FL}$ and pmCherry-V$_{1ab4}$ plasmids were obtained through Gibson assembly of the pmCherry backbone amplified by PCR with the primers Backbone-mCherry-fwd and Backbone-mCherry-rev (Supplementary Table 2), and the Vinculin$_{FL}$ and V$_{1ab4}$ inserts amplified with the primers Insert-Vinculin-fwd and Insert-Vinculin-rev (Supplementary Table 2), using vinculin$_{FL}$ and V$_{1ab4}$ cDNAs as templates respectively. The plasmid encoding EGFP-actin was a gift from B. Imhof (University of Geneva).

## Protein purification
V$_{FL}$, EGFP-Vh, V$_t$, V$_{1a}$, V$_{1ab4}$, T$_{FL}$, T$\Delta AI$, T$\Delta1$, T$\Delta2$, T$\Delta3$, T$\Delta1\Delta AI$, T$\Delta2\Delta AI$, T$\Delta3\Delta AI$, T$\Delta1\Delta AI\Delta ABD2$, T$_{F2F3R1R2R3}$, T$_{R2R3ABD3}$ and VBS$_1$ were expressed in *E.Coli* BL21 as previously described[38]. Briefly, 1 mM isopropyl β-D-1-thiogalactopyranoside (IPTG) was used for induction. Bacterial pellets of V$_{FL}$, V$_t$, V$_{1a}$, V$_{1ab4}$ were lysed by sonication in 20 mM Tris, pH 8.0, 1 M NaCl, 1 mM β-mercaptoethanol, 10 µg/ml benzamidine and 1 mM PMSF. Lysates of His-tagged vinculin constructs were purified by Ni-NTA-sepharose affinity chromatography (Ni$^{2+}$-nitrilotriacetic acid, Qiagen), followed by a Q-Sepharose ion exchange column. V$_t$ was purified as previously described[18]. Vinculin proteins were finally dialyzed in 20 mM Tris, pH 7.8, 1 mM DTT. Bacterial pellets of T$_{FL}$, T$\Delta AI$, T$\Delta1$, T$\Delta2$, T$\Delta3$, T$\Delta1\Delta AI$, T$\Delta2\Delta AI$, T$\Delta3\Delta AI$, T$\Delta1\Delta AI\Delta ABD2$, T$_{F2F3R1R2R3}$, T$_{R2R3ABD3}$ were lysed by sonication in 50 mM Tris, pH 7.8, 500 mM NaCl, 1% Triton X-100, 1 mM β-mercaptoethanol, 10 µg/ml benzamidine and 1 mM PMSF. Lysates of His-tagged talin constructs were purified by Ni-NTA-sepharose affinity chromatography (Ni$^{2+}$-nitrilotriacetic acid, Qiagen), followed by a gel filtration column (Superdex 200, 16/600, GE Healthcare). VBS$_1$ was purified as previously described[18]. Talin constructs were finally dialyzed in 20 mM Tris, pH 7.8, 100 mM KCl, 1 mM DTT.

## F-actin co-sedimentation assay
F-actin co-sedimentation assays were performed to determine the affinity of vinculin constructs for F-actin in the absence and presence of talin VBS$_1$. In F-actin co-sedimentation assay with vinculin alone, 2 µM of vinculin (V$_{FL}$, V$_t$, V$_{1a}$, V$_{1ab4}$) was incubated with 0, 1.5, 3, 4.5, 6, and 10 µM F-actin in 5 mM Tris, pH 7.8, 100 mM KCl, 1 mM MgCl$_2$, 0.2 mM EGTA, 200 µM ATP, 1 mM DTT during 15 min at room temperature. To test the effect of VBS$_1$, F-actin co-sedimentation assays were also performed in the presence of 2 µM of vinculin (V$_{FL}$, V$_{1a}$), 10 µM F-actin, with or without 1, 3, 5, 10 µM VBS$_1$ in 5 mM Tris, pH 7.8, 100 mM KCl, 1 mM MgCl$_2$, 0.2 mM EGTA, 200 µM ATP, 1 mM DTT during 15 min at room temperature. After centrifugation at 300,000 x g in a TLA-120.1 rotor (Beckman) during 30 min, the pellets and supernatants were separated and loaded on SDS-PAGE. Gels were scanned and analyzed with the ImageJ software.

## Polymerization assay
Actin polymerization was measured by the increase in fluorescence of 10 % pyrenyl-labeled actin in a SAFAS Xenius spectrofluorimeter (Safas, Monaco). To measure the elongation of actin filament barbed ends, actin polymerization was induced by adding 100 pM spectrin-actin seeds to 10% pyrenyl-labeled CaATP-G-actin in 5 mM Tris, pH 7.8, 100 mM KCl, 1 mM MgCl$_2$, 0.2 mM EGTA, 200 µM ATP, 1 mM DTT in presence of indicated proteins. The fraction of barbed end elongation was calculated as the ratio between the elongation rates in the presence and absence of proteins of interest. To test the ability of proteins to nucleate actin filaments, spontaneous polymerization was induced by adding 10% pyrenyl-labeled CaATP-G-actin in 5 mM Tris, pH 7.8, 25 mM KCl, 1 mM MgCl$_2$, 0.2 mM EGTA, 200 µM ATP, 1 mM DTT in presence of the proteins of interest or in 5 mM Tris, pH 7.0, 100 mM KCl, 1 mM MgCl$_2$, 0.2 mM EGTA, 200 µM ATP, 1 mM DTT at 37 °C.

## Observation of talin, vinculin, and single actin filaments in TIRF microscopy
Our protocol is a modification of protocols used to study talin and vinculin activities[18,22]. To prevent the nonspecific binding of actin filaments to the surface of the coverslip, we first irradiated coverslips with deep UVs for 3 min and incubated them with 0.1 mg/mL PLL-PEG for 1 h at room temperature. The coverslip was then washed extensively with water and dried. Flow cells containing 40-60 µl of liquid were prepared by sticking the PLL-PEG-coated coverslip to a slide with double-sided adhesive spacers. In experiments to observe and count single filament number, the chamber was first incubated with washing buffer (5 mM Tris, pH 7.8, 200 µM ATP, 1 mM DTT, 1 mM MgCl$_2$, 0.2 mM CaCl$_2$, 25 mM KCl) for 1 min. The chamber was then saturated with 10% BSA for 5 min and washed with washing buffer. The final reaction was then injected into the chamber. A typical reaction was composed of 1 µM actin (5% Alexa488-labeleld) in 5 mM Tris, pH 7.8, 200 µM ATP, 1% methylcellulose, 5 mM 1,4-diazabicyclo(2,2,2)-octane (DABCO), 25 or 100 mM KCl, 1 mM MgCl$_2$, 200 µM EGTA, 40 mM DTT supplemented with various talin and vinculin mutants. To observe talin, vinculin and actin simultaneously, 0.2 µM T$\Delta1\Delta AI$ (78% Alexa594-labeled) and 1 µM V$_{1ab4}$ (18% Alexa647-labeled) were first mixed, diluted 20 times to reach final concentrations of 10 nM T$\Delta1\Delta AI$ and 50 nM V$_{1ab4}$, injected on a flow chamber to immobilize T$\Delta1\Delta AI$-V$_{1ab4}$ complexes non-specifically on the surface passivated with PLL-PEG, and finally supplemented with 0.8 µM actin (5% Alexa488-labeled). Finally, we sealed the flow chamber with VALAP (a 1:1:1 mixture of vaseline, lanolin, and paraffin) and observed the reaction on a Nikon Eclipse Ti-E inverted microscope equipped with 60X/1.49NA objective (Nikon). The time-lapse videos were acquired by Metamorph and subsequently analyzed by the ImageJ software.

## Micropatterning
Coverslips were first cleaned by successive sonication in water and ethanol before being dried and then irradiated for 1 min under a UV

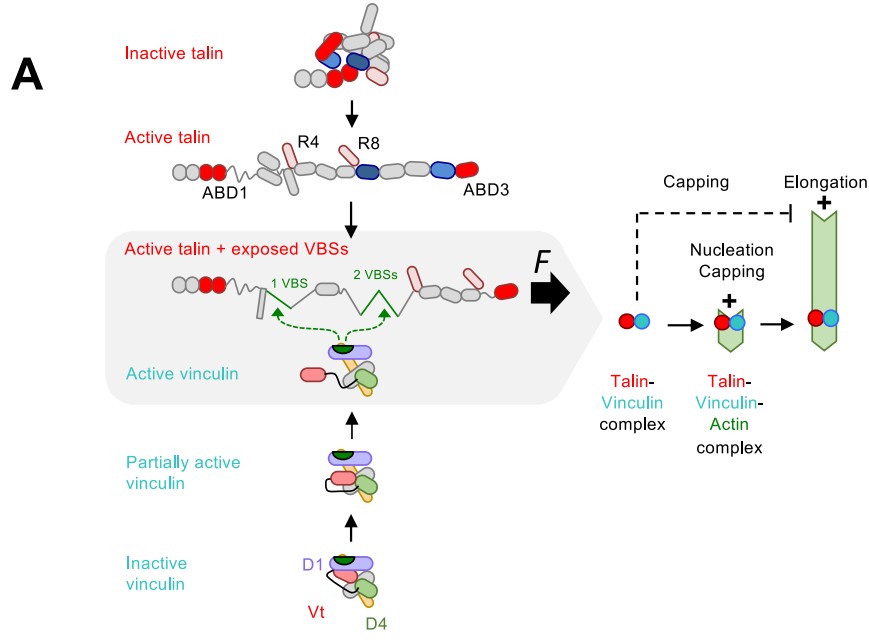

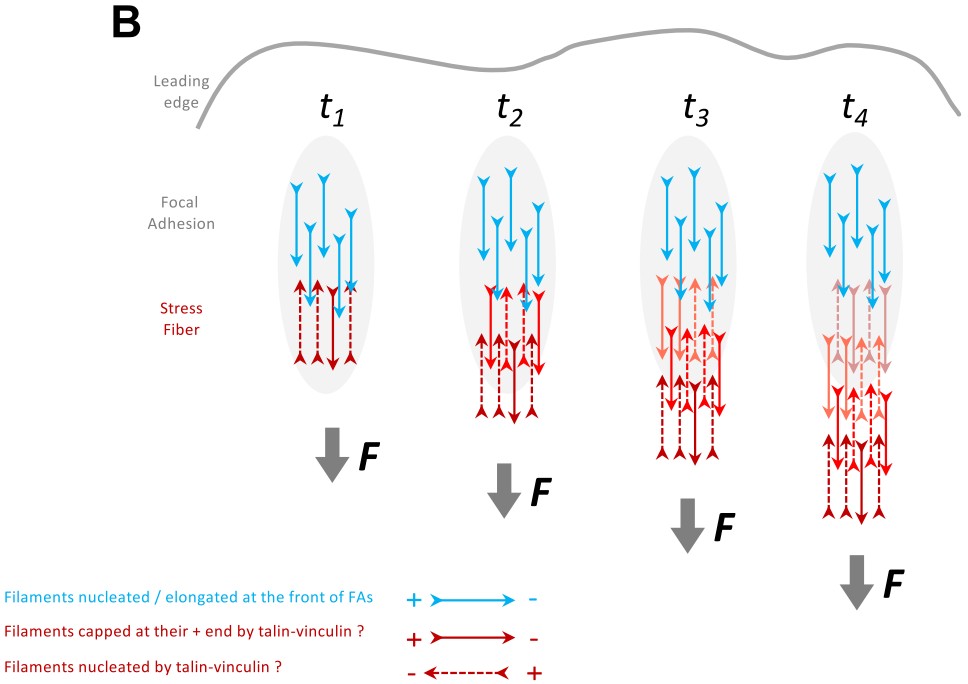

**Fig. 6 | Working model for the activities and regulation of talin, vinculin and the talin-vinculin complex. A** On the left side, talin (top) and vinculin (bottom) are shown according to their increasing degree of activation towards the central gray box in which the fully activated talin and vinculin form a complex. On the right side, a simplified representation shows the activity of the talin-vinculin complex nucleating an actin filament that is transiently capped at its barbed end before being released to elongate. **B** Scheme showing the known polarity of the actin filaments emerging from FAs by different colors depending on the orientation of their barbed end. The activities of the talin-vinculin complex that could explain these different orientations are indicated with the same color code.

lamp at 160 nm (Ossila). They were then incubated with a 0.1 mg/ml solution of PLL-PEG (SuSoS) for 2 h at room temperature. The coverslips were then placed with a drop of water on the washed and UV-activated chromium-quartz photomask, irradiated for 4 min, recovered and dried. Flow cells were prepared by sticking the PLL-PEG-coated coverslip to a slide with double-sided adhesive spacers. After addition of the proteins, the chamber was sealed with VALAP (a mixture of vaselin, lanolin and paraffin).

## Protein labeling

For 3-color imaging, actin was labeled with Alexa Fluor 488 carboxylic acid, succinimidyl ester, which reacts with the $NH_2$ groups of basic amino acids, as previously described[38]. T$\Delta$1$\Delta$AI and $V_{1ab4}$ were labeled on free cysteines with Alexa Fluor 594 maleimide and Alexa Fluor 647 maleimide respectively. Proteins were dialysed in 20 mM Tris HCl pH 7.8 and 100 mM KCl (T$\Delta$1$\Delta$AI) or 20 mM Tris HCl pH 7.8 ($V_{1ab4}$) overnight at 4 °C, then incubated for 3 hours with a 10-fold excess of Alexa

Fluor 594 maleimide (TΔ1ΔAI) or Alexa Fluor 647 maleimide (V$_{lab4}$). The reaction was then stopped with 1 mM DTT and dialysis was performed in 20 mM Tris HCl pH 7.8, 100 mM KCl and 1 mM DTT (TΔ1ΔAI) or 20 mM Tris HCl pH 7.8, 1 mM DTT (V$_{lab4}$), overnight at 4 °C and protected from light. Finally, the labeling ratio was calculated using the UV-visible absorbance spectrum and the molar extinction coefficients of the proteins and the dyes. The protein was then frozen in liquid nitrogen and stored at −80 °C.

### Cell lines and transfection

Hela cells (ATCC number CCL-2) were maintained in DMEM medium (Life Technologies) supplemented with 10% fetal bovine serum (FBS) (Biochrome) and 100 units/ml penicillin-streptomycin (anprotec) at 37 °C with 5% $CO_2$. 24 h before transfection, 300,000/well Hela cells were seeded in a 6-well plate. Hela cells were transfected using Fugene (Promega) and incubated for 48 h with pmCherry-Vinculin $_{FL}$ / pBFP-Talin $_{FL}$ or pmCherry-V$_{lab4}$ / pBFP-T$_{Δ1ΔAI}$ to be observed after fixation or, after additional transfection of pEGFP-actin, submitted to FRAP.

### Observation of fixed cells

Coverslips were incubated with 1 μg/cm² fibronectin (Merck) for 45 min. Transfected cells were seeded to fibronectin-coated coverslips. After 6 h, Hela cells were fixed in 4% paraformaldehyde in PBS (Santa Cruz) for 15 min at 37 °C and permeabilized for 10 min with 0.3% Triton X-100 (Thermofisher) in PBS. Fixed cells were blocked for 1 h with 5% FBS in PBS at 4 °C followed by 1 h staining with Alexa488-phalloidin (Thermofisher) and mounted in mounting media (Thermofisher). Images were acquired on an LSM 800 confocal laser scanning microscope (Zeiss) equipped with a 63×/1.4 NA oil objective with Airyscan.

### FRAP experiments

The FRAP experiment was performed with an LSM 800 confocal laser scanning microscope (Zeiss) equipped with a 63×/1.4 NA oil objective with Airyscan at 37 °C with 5% $CO_2$. EGFP-actin was bleached by a 10 times scan on the selected region with 10% intensity of 488 nm laser (10 mW). After bleaching, time-lapse images were acquired every 5 s. Images were analyzed by the Image J software (NIH).

### Statistics and reproducibility

Statistical analyses and graphical representations were carried out using GraphPad/Prism, Kaleidagraph, or Excel. Information on exact n values, statistical tests and their description, and error bars are given in the figure legends and the Source data file. No statistical method was used to predetermine sample size. No data were excluded from the analyses. The experiments were not randomized. The reproducibility is indicated in the figure legends.

### Reporting summary

Further information on research design is available in the Nature Portfolio Reporting Summary linked to this article.

## Data availability

Our manuscript is accompanied by a Source Data file providing the data used in Figs. 1B–E, 2B–E, 3E–H, 4E, and 5C, D, and Supplementary Figs. 2A–E, 3A–C, 6C–E, 7A–D, 8A–D, 9A–E, 10A–D, 11A–E, 12A–C, 13A–E, 14A–C, 15A–D, 16A–F, 17A, B, 18A, B, 19A–C, 20A, B, 21A, B, and 22C, D. Source data are provided with this paper.

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

## Acknowledgements

This work was supported by the Agence Nationale de la Recherche grants ANR-16-CE13-0007-02 PHAGOMECANO (C.L.C.), ANR-18-CE13-0026-01 RECAMECA (C.L.C.), ANR-21-CE13-0010-03 MYOCORTEX (C.L.C.), and ANR-22-CE44-0006-02 SwitchActin (C.L.C.). Funding to RG to support this project was received from Germany's Excellence Strategy EXC-2189, project ID 390939984 (R.G.). H.W. is supported by a PhD fellowship from the China Scholarship Council (CSC). R.S. is supported by a PhD fellowship from Association de Spécialisation et d'Orientation Scientifique. The present work has benefited from Imagerie-Gif core facility supported by Agence Nationale de la Recherche (ANR-10-INBS-04/FranceBioImaging; ANR-11-IDEX-0003-02/ Saclay Plant Sciences). We thank the members of the "Cytoskeleton Dynamics and Motility" team for helpful discussions.

## Author contributions

H.W. performed the binding assays, the kinetic assays, and the TIRF microscopy experiments on single actin filaments, the FRAP experiments in cells, purified most of the proteins, analyzed the data, and prepared the figures. R.S. performed some of the kinetic assays and TIRF microscopy experiments. C.N.V. performed binding assays on micro-patterned surfaces. V.H. designed and cloned some of the cDNAs. P.G. cloned cDNAs for cell studies. J.P. contributed to single filament microscopy assays. R.G. supervised cell studies. C.L.C. designed the experiments, supervised the project, and wrote the manuscript.

## Competing interests

The authors declare no conflicts of interest.

## Inclusion and ethics

The authors state that this work was carried out under conditions that ensure respect for and promotion of diversity and inclusion in science.
