## [Transparent Peer Review file · Nature Communications]

Talin and vinculin combine their activities to trigger actin assembly

Corresponding Author: Dr Christophe Le Clainche

Version 1:

Reviewer comments:

Reviewer #1

(Remarks to the Author)

Talin and vinculin are central proteins for actin stress fiber formation during the FA initiation. Upon activation, they form a complex and work together to interact with actin and induce actin bundling. However, because of their tight autoinhibition behavior, it has been challenging to elucidate the detail of the interactions of these two proteins and the combined action with actin. In this study, the authors applied biophysical analysis and attempt to understand protein function in terms of actin binding and nucleation using various deregulated talin and vinculin variants by introducing mutations that were derived from previous studies. Most of the experiments shown are confirmations or tests of previously shown or predicted interactions but the capping activities of vinculin/talin complex is interesting. However, I have several concerns about the observed activities and clarification would be necessary. I feel that the observations are too premature to support a publication with Nature Communications.

1. First, as the multi-modal activity of talin/vinculin is intriguing and relatively new, it would be essential that the authors verify these activities in a cellular context.
2. The authors showed that the talin-vinculin complex capped the barbed end of actin, and then to the pointed end, or got released from actin. What is the molecular mechanism of the recognition of those ends by talin/vinculin? It is a little confusing. The results would be less ambiguous if a competition assay with other capping proteins (such as formins, CapZ, Tropomodulin, gelsolin) can be performed.
3. Is there any sequence or domain similarity between talin/vinculin and the other end-binding proteins?
4. How do the authors explain the multiple talin-vinculin complex activities in the context of FA formation? The barbed end is near the plasma membrane where talin/vinculin are usually localized but the authors observed that the talin-vinculin would follow the pointed ends in their biophysical assays.

Reviewer #2

(Remarks to the Author)

In the article, Wang et al. investigate the combined activity of vinculin and talin on actin filament assembly, in a series of in vitro experiments. The authors designed numerous mutants of the two proteins to reproduce different levels of activation, without the need to apply forces. Mutants of vinculin and talin were then combined to assess their impact on 1. actin binding, 2. actin filament nucleation and 3. actin polymerization/capping. Based on these results, the authors propose that, in focal adhesions, vinculin and talin complexes actively participate in the nucleation, capture and capping of actin filaments.

Similar experiments have been performed by the authors (Le Clainche 2010, Ciobanasi 2018) with some active domains of vinculin and talin, and already showed that the proteins can nucleate and cap F-actin. The main contribution of this new article is to measure the cooperative activity of vinculin and talin at different activation levels. The authors also did a thorough job at testing the role of vinculin- and actin-binding domains of talin independently.

I think this article brings a significant contribution to our understanding of the role of vinculin and talin complexes in focal

adhesions, and I would support publications after the following revisions have been addressed.

Major points:

- I believe one could question the relevance of some results in more physiological conditions. One limitation concerns nucleation. As clearly stated in the manuscript, experiments are performed at low salt concentrations (25 mM KCl). It would be fair to remind the readers that this effect is weaker at higher salt concentrations, as shown by the authors in a previous publication where they looked at Vt only (Le Clainche 2020). I also suggest testing the nucleation at higher salt (e.g. 50 and 100 mM KCl) by at least one vinculin-talin complex (e.g. V1ab4 and TΔ1ΔA1).
- Moreover, in cells, most G-actin available for polymerization is in complex with profilin. I think the authors should test the impact of vinculin-talin (e.g. V1ab4 and TΔ1ΔA1) on the nucleation and polymerization of actin-profilin.
- I have some concerns about the capping experiments. Based on pyrene bulk assays only, it is difficult to tell whether the decrease in polymerization is due to barbed end capping or to other effects. First, vinculin-talin binding to the side of F-actin might reduce the pyrene fluorescence and thus estimated polymerization and capping. The authors should provide a control experiment showing the fluorescence intensity of fully polymerized actin, before and after the addition of vinculin-talin (vs buffer only). Or provide a reference.
- Moreover, in all plots, the elongation rate reaches a non-zero plateau at high vinculin concentrations. Instead of capping, vinculin-talin might just slow down the polymerization (like a leaky cap maybe). This can be solved by performing additional single filament experiments with vinculin and talin in solution (Fig 4).
- Concerning single filament experiments, the authors clearly characterized barbed end uncapping (Fig 4G and 5E). However, looking at kymographs and videos, it also seems that the polymerization rate is slower. The authors should provide a comparison of the polymerization rate with and without vinculin and talin. It should also be possible to compare the net elongation rates obtained with single filaments and in pyrene assays?

Maybe I missed something but I feel that the methods section is lacking quite some information. Here are a few points:

- What are the curves on all plots? Fits?
- There is no labelling protocol for V1ab4 and TΔ1ΔA1, or even a reference.
- Fig 4C-D: how was the density measured? Given the unit, I assume this is the number of filaments per surface area (and not the total length of filaments).
- I could not find the protocol for micropatterning (Fig 2B).
- Related to Fig 4: in single filament experiments, how are the pauses defined? E.g. is there a minimum pause duration? Can sticking to the surface cause pauses?
- F-actin cosedimentation assay: how are the gels analyzed? What is the formula to get the fraction of vinculin bound?

Minor points:

- Fig 1A: while the structure (and all the sketches throughout the figures) are extremely helpful, it is not clear which mutation is which.
- Fig 1B and Supp Fig 1: based on the gels, it seems that there is fewer VFL than V4 and V1b4 in pellets. However Fig 1B shows similar bound fractions. Is this normal?
- Fig 1E and 1G: discussed on page 7, the authors suggest that the difference might come from the F-actin concentration. Might it also be due to side vs barbed end binding?
- Fig 4A-B: at what time point were the images taken?
- Fig 4E: there is no definition for the arrowheads.
- Fig 5B: would it be possible to roughly estimate from the fluorescence signal the number of actin monomers between nucleation and uncapping?
- Supp Fig 3: "The control kinetics showing the polymerization of 2 μM actin alone are the same in all panels". B and C look different.
- Supp Fig 4: I got confused between the pink and red colors. I suggest having clearly distinct colors for ABD, AI and stars.
- Supp Fig 7A. VBS1 alone seems to accelerate polymerization. Can the authors comment on that?
- Supp Fig 8. Same comment, the addition of vinculin accelerates polymerization.

Version 2:

Reviewer comments:

Reviewer #1

(Remarks to the Author)

The authors sufficiently addressed my comments. I am happy to support their publication.

Reviewer #2

(Remarks to the Author)

I would like to acknowledge how seriously the authors took the revision process. They performed a number of new experiments to answer all comments raised by the other reviewer and I. I believe the data now strongly support the conclusions of the article.

I have one last minor comment: it seems that the significance tests are sometimes performed on individual measurements (e.g. Supp. Fig. 22). I would rather recommend to perform them on the average or median per experiment (as in Lord et al., JCB 2020, <https://doi.org/10.1083/jcb.202001064>).

I strongly support the publication of the revised manuscript to Nature Communications.

Point-by-point answers to the reviewers

Color code:

Black: Reviewers' comments

Blue: Our answers

Red: Citation of text added to revised manuscript

Reviewer 1

1. First, as the multi-modal activity of talin/vinculin is intriguing and relatively new, it would be essential that the authors verify these activities in a cellular context.

We have carried out several experiments in HeLa cells which, taken together, show that the binding of talin to vinculin stimulates actin assembly in focal adhesions. In summary, we show that expression of the constitutively binding mutants talin T Δ 1 Δ AI and vinculin V1ab4 leads to the thickening of stress fibers and the acceleration of their growth from focal adhesions. These results are the subject of the new Figure 5 and Supplementary Figure 22 and are described in the text p.11 as follows:

“The association of talin and vinculin contributes to the assembly of stress fibers in cells.

To determine the effect of talin-vinculin complex formation on actin assembly in cells, we compared HeLa cells co-expressing fluorescent wild-type talin and vinculin (BFP-Talin_{FL} and mCherry-Vinculin_{FL}) with cells co-expressing BFP-T Δ 1 Δ AI and mCherry-V_{1ab4} mutants. We first quantified FA area labelled with mCherry-vinculin and the width of stress fibers labelled with Alexa488-phalloidin. Cells expressing T Δ 1 Δ AI and V_{1ab4} have larger FAs associated with wider stress fibers (Supplementary Figure 22B, C, D), compared to cells expressing wild-type talin and vinculin (Supplementary Figure 22A, C, D).

To determine whether the constitutive formation of the talin-vinculin complex enhances stress fiber assembly dynamics, we compared the rate at which actin-GFP-containing stress fibers elongate by FRAP in HeLa cells co-expressing wild-type BFP-talin and mCherry-vinculin with cells co-expressing BFP-T Δ 1 Δ AI and mCherry-V_{1ab4} mutants (Figure 5A, B, Supplementary Movie 8, Supplementary Movie 9). As FAs slide retrogradely, we took this parameter into account and measured the rate of elongation of the stress fibers using the trailing edge of the FAs as a reference, revealed here by the fluorescence of mCherry-vinculin (Figure 5C). Quantification of the relative actin-GFP fluorescence recovery rate from the FAs indicates that stress fibers elongate faster in cells expressing T Δ 1 Δ AI and V_{1ab4} compared with cells expressing the wild-type talin and vinculin (Figure 5D).

Taken together, our data indicate that binding of vinculin to talin in FAs leads to the stimulation of actin assembly.”

2. The authors showed that the talin-vinculin complex capped the barbed end of actin, and then to the pointed end, or got released from actin. What is the molecular mechanism of the recognition of those ends by talin/vinculin? It is a little confusing. The results would be less ambiguous if a competition assay with other capping proteins (such as formins, CapZ, Tropomodulin, gelsolin) can be performed.

To demonstrate that the inhibition of actin polymerisation observed in kinetic assay under rapid elongation conditions, obtained by adding spectrin-actin seeds, reflects capping of the barbed end, we couldn't simply add another capping protein like CapZ or gelsolin, as replacing one capping protein with another would have had no measurable effect. Formins, as suggested by the reviewer, have the advantage of occupying the barbed end and accelerating its growth. It is therefore possible to determine whether the occupation of the barbed end by formins affects

the ability of the talin-vinculin complex to inhibit barbed end elongation. Our results show that the addition of the constitutive mutants of talin and vinculin ($T_{\Delta 1\Delta AI}$ and V_{1a}) is less effective in inhibiting the elongation of filaments whose barbed end is occupied by a fragment of the formin mDia1, than filaments whose barbed end is free to elongate, confirming that the talin-vinculin complex interacts with the barbed end. These results are the subject of the new Supplementary Figure 14 and are described in the text p.8 as follows:

“To confirm that this inhibition of elongation is indeed due to an interaction with the barbed end, we compared the ability of the $T_{\Delta 1\Delta AI}$ - V_{1a} complex to inhibit the elongation of spectrin-actin seeds and the elongation mediated by the FH1-FH2 fragment of the formin mDia1, which is known to elongate filaments while residing processively at their barbed end, thus protecting the filaments from barbed-end capping. Our quantifications show that actin filaments polymerizing in the presence of mDia1 are less effectively inhibited by the $T_{\Delta 1\Delta AI}$ - V_{1a} complex than filaments with free barbed ends, confirming the binding of the talin-vinculin complex to the barbed end (Supplementary Figure 14A-C).”

There is no evidence of capping of the pointed ends in our experiments observed by TIRF microscopy, because the pointed ends do not grow significantly at the actin concentration used, which is close to the critical concentration of the pointed end. The existence of a population of talin-vinculin complexes close to the pointed end (Figure 5E, former Figure 4E) means instead that these complexes were bound to the side or barbed end at the time of nucleation and remained associated with the side, near the pointed end, while the filament grows through its barbed end.

3. Is there any sequence or domain similarity between talin/vinculin and the other end-binding proteins?

To our knowledge, no protein has a domain that is truly homologous to the actin-binding tail of vinculin (Vt). However, it should be noted that some authors have reported similarity between the actin-binding proteins vinculin and alpha-catenin (Choi et al. 2012; Kobiela and Fuchs 2004). The fact that both proteins are capable of reducing the elongation of the barbed ends of actin filaments leaves open the possibility of an evolutionary conserved function of their actin-binding domains (Hansen et al. 2013, Le Clairche et al. 2010). For talin, two of the three actin-binding domains are conserved and found individually in other proteins, but the combination of the three actin-binding domains is a unique signature of talin. The N-terminal FERM domain of talin contains an actin filament binding domain that cap barbed ends (Ciobanasu et al. 2018). FERM domains are found in other actin-binding proteins, such as ERMs (Ezrin, Radixin, Moesin) and kindlins. However, ERMs interact with actin through a domain other than the FERM domain, called C-EMRAD (Senju and Tsai 2022), and none of ERMs and kindlins have been described as end-binding proteins. The C-terminal THATCH actin-binding domain of talin, also known as ILWEQ, is conserved in the mammalian endocytic proteins Hip1 and Hip1R, but has not been reported to target actin filament barbed ends (Gingras et al. 2007).

4. How do the authors explain the multiple talin-vinculin complex activities in the context of FA formation? The barbed end is near the plasma membrane where talin/vinculin are usually localized but the authors observed that the talin-vinculin would follow the pointed ends in their biophysical assays.

As explained above (end of paragraph 2.), the reason $T_{\Delta 1\Delta AI}$ and V_{1ab4} appear to be localized to the pointed end of single filaments in TIRF microscopy (Figure 5E, former Figure 4E) is that the pointed end elongates very slowly at 1 μ M actin, so no net growth of the pointed end is observed for the duration of the experiment, in contrast to the barbed end which elongates rapidly upon release from capping. The barbed ends of nucleated actin filaments are likely free to elongate in several possible directions, as soon as barbed end capping is released, which occurs with

a $t_{1/2} = 115 \text{ s} - 147 \text{ s}$ in our experimental conditions (Figure 3E, 4E). The ability of the complex to cap the barbed end of actin filaments can dictate the orientation of a subset of filaments whose barbed end faces the plasma membrane. Overall, our data are consistent with the mixed polarity of actin filaments associated with focal adhesions recently observed in cryoelectron tomography. These additional explanations are now part of the discussion on p14 as follows:

“The activities we have identified should generate filaments of different orientations in FAs. Actin filaments should be oriented with their barbed end facing the focal adhesion if the talin-vinculin complex caps their barbed end, or oriented with the barbed end towards the cell body or the leading edge if they have been nucleated and released. Although early electron microscopy studies concluded that the actin filaments making up the ends of stress fibers near the cell front are predominantly oriented with their barbed ends towards the membrane⁵¹, later studies benefiting from the precision of cryoelectron tomography have somehow revised these conclusions. These studies show that the stress fibers in contact with the front part of a focal adhesion contain filaments whose barbed ends are oriented towards the leading edge of the cell, and filaments of mixed polarities at the rear of the adhesion⁵², which is compatible with our findings (Figure 6B). The same orientations have been observed in platelet pseudopodia⁵³. The ability of the talin-vinculin complex to generate actin structures containing antiparallel filaments should promote contractility by myosin II.”

We admit that our previous Figure 6 showing single actin filaments elongating upwards misled the reader. For this reason, we have added a new panel in the final figure that takes into account all our results in the context of focal adhesion (Figure 6B).

Reviewer 2

1. I believe one could question the relevance of some results in more physiological conditions. One limitation concerns nucleation. As clearly stated in the manuscript, experiments are performed at low salt concentrations (25 mM KCl). It would be fair to remind the readers that this effect is weaker at higher salt concentrations, as shown by the authors in a previous publication where they looked at Vt only (Le Clairche 2010). I also suggest testing the nucleation at higher salt (e.g. 50 and 100 mM KCl) by at least one vinculin-talin complex (e.g. V1ab4 and T Δ 1 Δ A1).

We performed actin polymerization kinetics at 25, 50, 75 and 100 mM KCl in the presence of V_{1ab4} and T Δ 1 Δ A1 as requested by the reviewer. Results expressed as elongation rate versus KCl concentration show that nucleation activity is strong at 25 mM KCl and remains measurable at 50 mM KCl. At 75 mM KCl and above, nucleation activity is no longer measurable in our conditions (20°C, pH 7.8). This series of experiments is shown in supplementary figure 17 and the results are described in the Results section p9 as follows:

“This nucleation activity is strong at 25 mM KCl and remains significant at 50 mM KCl but disappears at 75 mM KCl, indicating the electrostatic nature of interactions between talin, vinculin and actin (Supplementary Figure 17A-B).”

Although the nucleation activity observed in the presence of V_{1ab4} and T Δ 1 Δ A1 is lost at 100 mM KCl, it should be noted that the conditions we used (pH=7.8, 20°C), which are the conditions typically used in the field, are also far from the physiological conditions of the cytoplasm of a living cell. This is why we carried out experiments at pH=7 and 37°C, which are closer to cellular conditions. Under these conditions (37°C, pH=7, 100 mM KCl), we restored nucleation by V_{1ab4} and of T Δ 1 Δ A1. This series of experiments is shown in supplementary figure 18 and the results are described in the Results section p9 as follows:

“The conditions we used to measure the stimulation of actin assembly by talin and vinculin are close to those generally used to study actin polymerization activities in vitro with purified proteins. However, these conditions (20°C, pH=7.8, 25 mM KCl) are far from physiological conditions. We therefore performed kinetic assays under physico-chemical conditions as close as possible to those found in mammalian cells, i.e. 37°C, pH=7, 100 mM KCl. Under these conditions, spontaneous actin polymerization is faster and the stimulation of actin polymerization by the combined action of T_{Δ1ΔA1} or T_{Δ3ΔA1} and V_{1ab4} is strong (Supplementary Figure 18A-B).”

2. Moreover, in cells, most G-actin available for polymerization is in complex with profilin. I think the authors should test the impact of vinculin-talin (e.g. V1ab4 and TΔ1ΔA1) on the nucleation and polymerization of actin-profilin.

We tested the effect of the combination of V_{1ab4} and T_{Δ1ΔA1} on actin polymerization in the presence of profilin. This series of experiments is shown in supplementary figure 18 and the results are described in the Results section p9 as follows:

“Importantly, in cells, polymerizable actin is complexed with profilin, which prevents spontaneous nucleation of actin filaments and restricts the formation of new filaments at specific sites, generally associated with a membrane-bound structure stimulated by chemical or mechanical signals. We therefore tested the effect of profilin on nucleation activity. Although profilin does not completely abolish this activity, the very strong reduction observed suggests that an additional mechanism is involved in the dissociation of profilin (Supplementary Figure 18B).”

We have added a paragraph on the link between actin nucleator activity and profilin in the discussion section on p14 as follows:

“The ability of a newly discovered nucleation machinery to form filaments in the presence of profilin has always been the subject of debate. Profilin prevents spontaneous polymerization of actin filaments in the cytoplasm where they would not play a role in producing force against the plasma membrane⁴. The generally accepted idea is that nucleation systems, located at the membrane and under the control of specific signaling pathways, trigger the polymerization of actin complexed with profilin. However, few nucleators are capable of nucleating actin filaments using profilin-actin complexes alone. Thus, the nucleation of branched filaments by the Arp2/3 complex or the nucleation of linear filaments by formins are severely inhibited in the presence of high profilin concentrations^{48,49}, as observed for the talin-vinculin complex. It is likely that a pool of free actin, resulting from the constant depolymerization of actin networks, is transiently available to feed nucleators such as the talin-vinculin machinery that we described, before it associates to profilin. In the cell, the concentration of profilin is lower than that of actin⁵⁰. Actin is therefore partially free, or bound to other proteins, such as thymosin β₄, which sequesters actin monomers, or to other proteins that could be compatible with nucleators.”

3. I have some concerns about the capping experiments. Based on pyrene bulk assays only, it is difficult to tell whether the decrease in polymerization is due to barbed end capping or to other effects. First, vinculin-talin binding to the side of F-actin might reduce the pyrene fluorescence and thus estimated polymerization and capping. The authors should provide a control experiment showing the fluorescence intensity of fully polymerized actin, before and after the addition of vinculin-talin (vs buffer only). Or provide a reference.

Our multiple observations that the addition of V_{1ab4} and T_{Δ1ΔA1} increases pyrene actin fluorescence in the absence of spectrin-actin seeds (Figure 2D, E), indicate that these proteins do not quench pyrene actin fluorescence. These observations allow us to confidently interpret

a decrease in fluorescence in the presence of spectrin-actin seeds as an inhibition (Figure 2B, C). To remove any ambiguity, these pyrene actin polymerization kinetics were supplemented with TIRF microscopy experiments showing that single filaments exhibit pauses in the presence of V_{1ab4} and $T_{\Delta 1\Delta A1}$ that are unambiguous signatures of capping of the barbed end (Figure 3).

4. Moreover, in all plots, the elongation rate reaches a non-zero plateau at high vinculin concentrations. Instead of capping, vinculin-talin might just slow down the polymerization (like a leaky capter maybe). This can be solved by performing additional single filament experiments with vinculin and talin in solution (Fig 4).

First it is important to remind the reviewer that this partial effect is obtained in the presence of a fixed concentration of V_{1a} (2 μ M) and increasing concentration of $T_{\Delta 1\Delta A1}$, which means that the effect is limited by the concentration of V_{1a} . It was difficult to vary all the parameters to reach the saturation of actin filaments barbed ends. Therefore, single filament observation is a good complement as suggested. Complete block of single filament elongation, reflecting capping events, have been observed in Figure 3 (previously Figure 4 mentioned by the reviewer) and Figure 4 (previously Figure 5). However, we provide a new observation in TIRF microscopy at 100 mM KCl, which are the conditions used in Figure 2 (previously Figure 3) that was missing in Figure 3 (previously Figure 4 mentioned by the reviewer), showing the presence of pauses in the elongation of single filaments in the presence of $T_{\Delta 1\Delta A1}$ and V_{1ab4} . The results are presented in Figure 3 and described in the results section on p10 as follows.

“At high salt concentration (100 mM KCl), which restricts the activity of the talin-vinculin complex to the capping of the barbed ends only (Figure 2B, C), the filaments show pauses interrupting the elongation of their barbed end (Figure 3A, D, Supplementary Movie 1). Quantifying the distribution of these pause times enabled us to estimate a dissociation rate constant of the complex from the barbed of 0.006 s^{-1} ($t_{1/2} = 115.8 \text{ s}$) (Figure 3E). Quantification of the elongation rate between pauses revealed a slight reduction in the presence of V_{1ab4} and $T_{\Delta 1\Delta A1}$ reflecting the presence of remaining short capping events that escaped our analysis or an effect on the elongation other than barbed end capping (Figure 3F).”

5. Concerning single filament experiments, the authors clearly characterized barbed end uncapping (Fig 4G and 5E). However, looking at kymographs and videos, it also seems that the polymerization rate is slower. The authors should provide a comparison of the polymerization rate with and without vinculin and talin. It should also be possible to compare the net elongation rates obtained with single filaments and in pyrene assays?

It is true that there is a capping effect and also an effect on the elongation rate between capping events. In this revised version, we provide a quantification of the elongation rates between pauses measured on single filaments observed in TIRF microscopy and compared them with actin alone, as well as a possible explanation.

The effect on the elongation rate between pauses is observed at 100 mM KCl as described on p10:

“Quantification of the elongation rate between pauses revealed a slight reduction in the presence of V_{1ab4} and $T_{\Delta 1\Delta A1}$ reflecting the presence of remaining short capping events that escaped our analysis or an effect on the elongation other than barbed end capping (Figure 3F).”

No effect on the elongation rate between pauses is observed at 25 mM KCl as described on p10:

“Quantification of the elongation rate between pauses did not reveal any effect of V_{1ab4} and $T\Delta 1\Delta AI$ (Figure 3F).”

- What are the curves on all plots? Fits?

The curves shown in the graphs of Figure 3E and Figure 4E are exponential decay fits that allow us to extract a $t_{1/2}$ or an apparent dissociation rate constant (in s^{-1}) for the proteins talin and vinculin from the barbed end of an actin filament as described in Le Clainche et al. 2010. In contrast, the curves appearing in the graphs of Figure 1B-D, Figure 2B-E, Supp. Figure 6D, Supp. Figure 13E, Supp. Figure 14C, Supp. Figure 16F, Supp. Figure 17B do not correspond to the fits of the experimental points by an equation corresponding to a detailed mechanism. Indeed, formalization of the mechanism, involving a complex stoichiometry of talin and vinculin that interact with each other and with actin via several actin-binding domains, seems out of reach. The different constants (K_d) are so numerous that it would be easy to fit the experimental points but the value of such a fit would be almost null due to the absence of constraints offered by the multiple parameters left free. The curves are therefore not fits of the experimental points by a model, but are drawn manually to reflect the trend of the data, in order to make it easier to read the data. These explanations have been added to the figure legends.

This note has been added at the end of the legends of Figures 1 and 2:

“In (B-E), the curves are not fits of the experimental points by a model, but are drawn manually to reflect the trend of the data, in order to make the data easier to read.”

- There is no labelling protocol for V_{1ab4} and $T\Delta 1\Delta AI$, or even a reference.

We thank the reviewer for pointing out the absence of this protocol, which is the subject of a paragraph on p17 in the materials and methods section of this revised version as follows:

“Proteins were dialysed in 20 mM Tris HCl pH 7.8 and 100 mM KCl ($T\Delta 1\Delta AI$) or 20 mM Tris HCl pH 7.8 (V_{1ab4}) overnight at 4°C, then incubated for 3 hours with a 10-fold excess of Alexa Fluor 594 maleimide ($T\Delta 1\Delta AI$) or Alexa Fluor 647 maleimide (V_{1ab4}). The reaction was then stopped with 1 mM DTT and dialysis was performed in 20 mM Tris HCl pH 7.8, 100 mM KCl and 1 mM DTT ($T\Delta 1\Delta AI$) or 20 mM Tris HCl pH 7.8, 1 mM DTT (V_{1ab4}), overnight at 4°C and protected from light. Finally, the labelling ratio was calculated using the UV-visible absorbance spectrum and the molar extinction coefficients of the proteins and the dyes. The protein was then frozen in liquid nitrogen and stored at -80°C.”

- Fig 4C-D: how was the density measured? Given the unit, I assume this is the number of filaments per surface area (and not the total length of filaments).

Filament density is defined as the number of filaments per unit area (number of filaments / μm^2).

- I could not find the protocol for micropatterning (Fig 2B).

We thank the reviewer for pointing out the absence of this protocol, which is the subject of a paragraph on p17 in the materials and methods section of this revised version. Note that this protocol is presented briefly because we have already published it in Vigouroux et al. 2020. Figure 2B is now Supplementary Figure 6B in this revised version. The paragraph on p17 is:

“Coverslips were first cleaned by successive sonication in water and ethanol before being dried and then irradiated for 1 min under a UV lamp at 160 nm (Ossila). They were then incubated with a 0.1 mg/ml solution of PLL-PEG (SuSoS) for 2 h at room temperature. The coverslips

were then placed with a drop of water on the washed and UV-activated chromium-quartz photomask, irradiated for 4 min, recovered and dried. Flow cells were prepared by sticking the PLL-PEG-coated coverslip to a slide with double-sided adhesive spacers. After addition of the proteins, the chamber was sealed with VALAP (a mixture of vaselin, lanolin and paraffin).”

- Related to Fig 4: in single filament experiments, how are the pauses defined? E.g. is there a minimum pause duration?

The minimum duration of a pause is the time interval between two images in a time lapse, which is 10 seconds. The accuracy of the simple exponential fit does not indicate any over- or under-representation of the shortest pauses (Figure 3, former Figure 4).

- Can sticking to the surface cause pauses?

It is true that the effects of filament interaction and friction with the surface has been cited as a problem affecting the conclusions of single filament experiments in TIRF. However, in a previous work we immobilized biotin-labeled actin filaments on a PEG-biotin surface, comparable to the PEG surface used here, through streptavidin, and found no significant artefactual effect (Ciobanasu et al. 2018). Furthermore, the fact that kinetic studies in solution and single filament observations were conducted in parallel with the same conclusions rules out the possibility of conclusions drawn from artefactual observations.

- F-actin cosedimentation assay: how are the gels analyzed? What is the formula to get the fraction of vinculin bound?

First, we would like to point out that we repeated the cosedimentation experiments in the article to improve the statistics, which led to the decision to reduce the number of mutants tested in Figure 1 to those used in the rest of the article (see last paragraph).

To answer the question precisely, in this revised version of our manuscript, the fraction of bound vinculin was calculated as the amount of vinculin in the pellet divided by the sum of vinculin in the pellet and supernatant.

Minor points

- Fig 1A: while the structure (and all the sketches throughout the figures) are extremely helpful, it is not clear which mutation is which.

We admit that the position of the mutations on the structure in Figure 1A was very approximate. To remove any ambiguity in reading this figure, we have indicated the modified amino acids with a color code and greater precision.

- Fig 1B and Supp Fig 1: based on the gels, it seems that there is fewer VFL than V4 and V1b4 in pellets. However Fig 1B shows similar bound fractions. Is this normal?

As indicated above, we repeated the cosedimentation experiments and reduced the mutants tested to V1ab4 and V1a. The gels shown are those from the new, cleaner experiments and correspond well to the quantifications presented in Figure 1.

- In Supp Fig1, the actin band in VFL is smaller than V4 and V1b4.

As mentioned above, we performed new cosedimentation assays. In the new Supplementary Figure 1A-C, the actin bands have the same intensity and size in all conditions.

- Fig 1E and 1G: discussed on page 7, the authors suggest that the difference might come from the F-actin concentration. Might it also be due to side vs barbed end binding?

In summary, vinculin V1ab4 binds to actin filaments in a cosedimentation assay containing up to 10 μM of pre-polymerized actin filaments, with free barbed ends at an unknown concentration. V1ab4 inhibits the elongation of actin filaments in a kinetic assay containing 2 μM of total actin which polymerizes very rapidly thanks to the addition of spectrin-actin seeds providing 100 pM of free barbed ends. In contrast, V1ab4 did not stimulate significantly the polymerization of 1.5 μM of monomeric actin in a kinetic assay. We can therefore hypothesize that the actin filaments contribute to the complete activation of vinculin. However, our experiments do not allow us to propose a specific role for the barbed ends of the actin filaments in this mechanism.

- Fig 4A-B: at what time point were the images taken?

Images were taken at 790 s as now indicated in the revised version (Figure 3A-C).

- Fig 4E: there is no definition for the arrowheads.

The arrowheads correspond to capping and uncapping events. We have added this missing information in the legend of the figure (Figure 3D in this revised version).

- Fig 5B: would it be possible to roughly estimate from the fluorescence signal the number of actin monomers between nucleation and uncapping?

Considering that the total intensity of a portion of filament of 1 μm length corresponds to 400 actin subunits, we can estimate that there are about 10 actin subunits in the actin spot that corresponds to a nucleation event stopped before elongation (estimation from Figure 4B, former Figure 5B).

- Supp Fig 3: “The control kinetics showing the polymerization of 2 μM actin alone are the same in all panels”. B and C look different.

We thank the reviewer for pointing out these differences. In fact, the control curves were the same for the experiments carried out on the same day. As some experiments were performed on a different day, their control curves were different.

In this revised version of the manuscript, we have decided to simplify some of our figures. In particular, we have removed several mutants that were not used later in the manuscript in Figure 1. As a result, Supplementary Figure 3 contains only 3 panels called A, B and C corresponding to the previous A, B and F. The control curves are the same for the panels A and C performed the same day while a different control curve is used for B performed on a different day, as indicated in the legend.

- Supp Fig 4: I got confused between the pink and red colors. I suggest having clearly distinct colors for ABD, AI and stars.

We have changed the color code of this figure to avoid confusion. In this revised version ABDs are in pink, AIs in blue, VBSs in green and VBS-exposing mutations appear as green stars.

- Supp Fig 7A. VBS1 alone seems to accelerate polymerization. Can the authors comment on that?

The quantification of the slopes in the previous Figure 1G showed that this eye-catching effect observed in the kinetics presented in the previous Supplementary figures 7A was real but negligible. This supplementary Figure 7 and the corresponding Figure 1G have been removed in this revised version to simplify the article as several mutants were not used later in the paper.

- Supp Fig 8. Same comment, the addition of vinculin accelerates polymerization. Small effect on the slope...

The quantification of the slopes in the supplementary Figure 6D (previously Figure 2D) shows that this small effect of vinculin on actin polymerization observed in the kinetics presented in the Supplementary Figure 7 (previously Supplementary figures 8) is negligible. Supplementary Figure 6D shows that there is no major difference between the points corresponding to V1a without talin (first points at 0 μ M TFL, T Δ 1, T Δ 2, T Δ 3) and the points without V1a and without talin (first points at 0 μ M TFL, T Δ 1, T Δ 2, T Δ 3).

Justification for additional changes not requested by reviewers

We decided to reduce the number of mutants tested in Figure 1 of this revised version, although this was not requested by the reviewers. We believe that Figure 1, which is the only figure affected by this change, was difficult to understand in the previous version because the 6 vinculin mutants, plus the Vt domain and the full-length VFL protein, were not used in all experiments. We also simply recognize that performing all possible tests with all these mutants would be beyond the capabilities of our team. The decision to remove the data for V4, V1ab, V1a4, V1b4 also seemed reasonable to us as the important activities were contributed by V1a, and V1ab4 which were sufficient to study capping and nucleation activities when combined with talin constructs, whereas the other vinculin mutants were redundant or simply less useful and not used in the rest of the study. It should be added that the work on the mutants V4, V1ab, V1a4, V1b4 that we removed was a preliminary study leading to the V1ab4 mutant, which combines all the mutations. In summary, we have removed 4 mutants (V4, V1ab, V1a4, V1b4) from Figure 1 but have not changed any of the talin mutants used throughout the paper. A final reason for this choice was to focus on physiological relevance rather than mechanistic details, as we understood from the reviewers' comments that this was the priority. We are therefore convinced that deleting these data does not alter the message of the article and, on the contrary, makes it clearer.

Point-by-point answers to the reviewers

Color code:

Black: reviewers' comments

Blue: our answers

Reviewer #1

The authors sufficiently addressed my comments. I am happy to support their publication.

We thank reviewer 1 for supporting the publication of our article.

Reviewer #2

I would like to acknowledge how seriously the authors took the revision process. They performed a number of new experiments to answer all comments raised by the other reviewer and I. I believe the data now strongly support the conclusions of the article.

I have one last minor comment: it seems that the significance tests are sometimes performed on individual measurements (e.g. Supp. Fig. 22). I would rather recommend to perform them on the average or median per experiment (as in Lord et al., JCB2020, <https://doi.org/10.1083/jcb.202001064>).

I strongly support the publication of the revised manuscript to Nature Communications.

Regarding the minor comment on the significance tests, we have modified the Supplementary Figures 22C and Supplementary Figures 22D to show all experimental points in 3 different colors corresponding to the three independent experiments. The means of the 3 experiments are shown and the statistical tests have been performed on these means. The other data in our study are not suitable for this treatment.

We thank reviewer 2 for acknowledging the work we have done to meet the requests made by reviewers 1 and 2 and for strongly supporting the publication of our paper.